# Identify Critical KV Cache in LLM Inference from an Output Perturbation Perspective

## Abstract

Large language models have revolutionized natural language processing but face significant challenges of high storage and runtime costs, due to the transformer architecture's reliance on self-attention, particularly the large Key-Value (KV) cache for long-sequence inference. Recent efforts to reduce KV cache size by pruning less critical entries based on attention weights remain empirical and lack formal grounding. This paper presents a formal study on identifying critical KV cache entries by analyzing attention output perturbation. Our analysis reveals that, beyond attention weights, the value states within KV entries and pretrained parameter matrices are also crucial. Based on this, we propose a perturbation-constrained selection algorithm that optimizes the worst-case output perturbation to identify critical entries. We demonstrate that our algorithm is a universal, plug-and-play enhancement that incurs negligible computational overhead. When integrated with three state-of-the-art cache eviction methods on three distinct LLMs, our algorithm significantly reduces the compression loss by more than *half* on average across 29 datasets from the Ruler and LongBench benchmarks. Further perturbation analysis, at both the head and layer levels, confirms the principles underlying our effectiveness. This work offers a new, formally grounded perspective to the cache eviction field, opening promising avenues for future research.

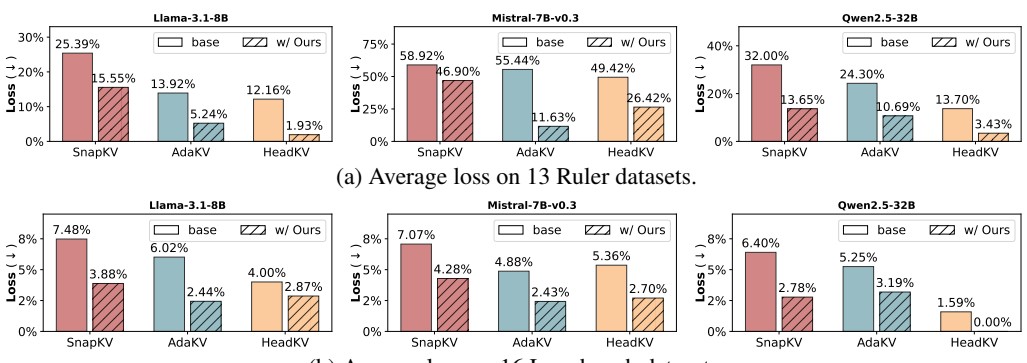

(a) Average loss on 13 Ruler datasets.

(b) Average loss on 16 Longbench datasets.

Figure 1: Our algorithm reduces the loss of three existing cache eviction methods by more than *half* on average. (shown at 40% cache size; see experiments for other sizes).

## 1 Introduction

Large language models (LLMs) using transformer architecture have excelled in many tasks, likes chatbots (Achiam et al., 2023)(Yi et al., 2024) and intelligent agents (Wang et al., 2024). However, the quadratic computational cost inherent in the transformer's self-attention mechanism poses significant challenges for practical deployment. To mitigate this, LLMs often use a Key-Value (KV) cache, which stores intermediate results from the self-attention mechanism. Each KV cache entry corresponds to the KV states of a past token, thus allowing for the bypassing of recomputation of these tokens during autoregressive generation. However, as sequence lengths increase, the number of KV cache entries expands dramatically. This expansion not only leads to considerable GPU memory overhead but also significantly increases I/O latency, hindering the real-world deployment (Sun et al., 2024a).

Recent research has identified that only a subset of KV cache entries substantially contribute to the output of the self-attention mechanism (Zhang et al., 2024b; Liu et al., 2024b; Tang et al., 2024a). As

a result, many methods, known as *cache eviction*, have been developed to reduce the KV cache size to fit within a given budget by evicting non-critical entries during inference. These methods effectively save GPU memory and improve subsequent decoding speed. Notably, H2O (Zhang et al., 2024b) and Scissorhands (Liu et al., 2024b) observe a power-law distribution of attention weights: a small fraction of KV cache entries consistently dominates the majority of attention weights, aligning closely with the concept of cache entry criticality during inference. These methods introduce frameworks that leverage accumulated attention weights to identify and preserve critical cache entries. Building on this, subsequent works (Adnan et al., 2024; Li et al., 2024; Feng et al., 2024; Fu et al., 2024) have refined attention weight accumulation and added operations like pooling and budget allocation to better preserve key information. However, while these methods generally assume that entries with higher attention weights—determined by the similarity between key states in the KV cache and the target query state—are critical, the identification and characterization of "critical cache entries" remain unformalized. This assumption raises two key questions:

1. *What criteria determine the critical KV cache?*
2. *Is reliance on attention weights alone sufficient for identifying critical cache entries?*

In this paper, we define the problem of critical cache identification from the perspective of output perturbation. This approach is grounded in the key insight that KV cache eviction loss is driven by changes in the attention output. Our primary objective, therefore, is to minimize this perturbation when replacing the full KV cache with only its critical entries. To formalize this, we introduce a theoretical framework that bounds the worst-case perturbation to guide practical optimization. Specifically, to quantify this perturbation, we employ the simple $L_1$ distance and derive its upper bound[1], corresponding to the worst-case perturbation. Our analysis reveals that this upper bound is influenced by both the attention weights and the value states projected through the parameter matrix. Based on these insights, we propose a perturbation-constrained selection algorithm designed to minimize this derived upper bound. It goes beyond mere reliance on attention weights, underscoring the significance of previously overlooked value states and the pretrained parameter matrix.

We integrate our algorithm into three state-of-the-art (SOTA) cache eviction methods, SnapKV (Li et al., 2024), AdaKV (Feng et al., 2024) and HeadKV (Fu et al., 2024), replacing their reliance on solely attention-weight-based strategies. Comprehensive evaluations on 29 datasets from Ruler and LongBench, as summarized in Figure 1, demonstrate that our method serves as a universal enhancement, substantially improving post-eviction generation quality. Further empirical analysis confirms and elucidates the practical benefits of our algorithm: (1) It effectively reduces output perturbation in over 92% of the Llama model's attention heads. (2) Its advantages accumulate across layers, significantly lowering the perturbation in final-layer hidden states. (3) It consistently performs well across various cache sizes, robustly mitigating quality loss under different resource constraints in practical applications. Our contributions can be summarized as follows:

1. We highlight that current cache eviction methods neglect the crucial problem of identifying critical KV cache entries. To address this, we propose using output perturbation as a criterion for determining criticality. Our analysis shows that attention weights alone are insufficient; the value states projected by the parameter matrix are also essential.

2. Building on the constraint of worst-case output perturbation, we propose a novel critical entry selection algorithm as a universal enhancement. When integrated with three SOTA eviction techniques, it reduces compression loss by more than half on average, as validated across three distinct LLMs on 29 datasets from Ruler and LongBench benchmarks (Figure 1).

3. Further empirical analysis examines and confirms the benefits of our perturbation-constrained selection algorithm. This analysis also highlights the significant potential for optimizing critical cache selection from the theoretical perspective of output perturbation.

## 2 RELATED WORKS

**Perturbation-based analysis** has achieved remarkable success in neural network interpretability and pruning. For example, Catformer (Davis et al., 2021) and Admin (Liu et al., 2020) utilize output

---

[1]We choose the $L_1$ distance for its simplicity and effectiveness, though more complex metrics are also compatible with our framework. For example, employing the $L_2$ distance yields similar gains (see Appendix J).

perturbation analysis to create more stable network architectures and enhance training methods. Similarly, pruning techniques (Han et al., 2015; Frantar & Alistarh, 2023), with Wanda (Sun et al., 2024b) as a representative, aim to identify neurons whose removal minimally impacts output, thereby reducing network parameters. In this paper, we present the first analysis of output perturbations aimed at developing more effective selection metrics for cache eviction in efficient LLM inference.

**KV cache eviction** aims to retain only critical KV cache entries while evicting non-essential ones to reduce cache size, facilitating efficient long-sequence inference in LLMs. Early methods (Xiao et al., 2023), which preserved recent entries in a sliding window, risked losing important information in long sequences. Techniques like H2O (Zhang et al., 2024b) and Scissorhands (Liu et al., 2024b) used accumulated attention scores to identify key entries, aiming to retain crucial context. Subsequent works refined these methods (Ge et al., 2024b; Adnan et al., 2024; Ge et al., 2024a; Li et al., 2024), with SnapKV (Li et al., 2024) achieving the SOTA performance through introducing observation window-based attention weight accumulation and pooling operations. However, these methods are largely empirical, relying solely on attention weights to identify critical entries. Our paper introduces a novel perturbation-constrained selection algorithm based on in-depth analysis from an output perturbation perspective. This algorithm seamlessly integrates into existing cache eviction methods without altering underlying accumulation processes. Additionally, recent advances in budget allocation optimization (Yang et al., 2024a; Zhang et al., 2024a; Fu et al., 2024; Xiao et al., 2024a; Zhang et al., 2025b)—such as AdaKV [(Feng et al., 2024)], which adaptively allocating budgets based on head characteristics, and HeadKV (Fu et al., 2024), which uses fine-grained offline profiling to guide allocation—are fundamentally orthogonal to our approach. To comprehensively demonstrate the effectiveness, we integrate our algorithm with these three representative cache-eviction lines—SnapKV, AdaKV, and HeadKV—and observe substantial gains across all three.

## 3 CRITICAL KV CACHE ENTRY SELECTION

For critical cache entry selection, we aim to choose cache entries that represent the entire KV cache during self-attention computation, producing an output that is a close approximation, if not identical. Base on this insight, we formalize the problem of identifying critical cache entries from the perspective of output perturbation (Definition 3.1) in Section 3.2. Subsequently, in Section 3.3, we formalize the output perturbation and derive its upper bound. Then, we propose a two-stage greedy algorithm in Section 3.4 that constrains worst-case perturbations for selecting critical entries, with theoretical analysis provided in Section 3.5. Finally, in Section 3.6 we integrate the algorithm into current SOTA cache eviction methods.

### 3.1 PRELIMINARIES

LLMs utilizing the multi-head self-attention mechanism operate with an autoregressive generation approach. In this setup, each decoding step leverages the most recently generated token to predict the next one. To illustrate this process, we focus on a single attention head as an example. Let $X \in \mathbb{R}^{n \times d}$ denote the embedding matrix for all tokens in the sequence, with $x = X_{-1,:} \in \mathbb{R}^{1 \times d}$ representing the embedding vector of the most recent token, which serves as input at the current time step. The parameter matrices, denoted by $W^Q$, $W^K$, and $W^V \in \mathbb{R}^{d \times d_h}$ are used to map the token embeddings into their respective Query, Key, and Value states with head dimension $d_h$ as follows:

$$q = xW^Q; K = XW^K; V = XW^V \tag{1}$$

During the decoding phase, the Key and Value states of previously generated tokens (represented by $X$) are stored in the KV cache, allowing for the elimination of redundant computation. Accordingly, the query $q$, derived from the most recent token $x$, attends to the cached Key $K$ to compute the attention weights $A$. These weights are then applied to the cached Value $V$, producing an intermediate output. This intermediate result is subsequently transformed into the final output $o$ of the self-attention mechanism by the output parameter matrix $W^O \in \mathbb{R}^{d_h \times d}$:

$$o = AVW^O, \text{ where } A = \text{softmax}\left(qK^T/\sqrt{d}\right) \tag{2}$$

### 3.2 WHAT CRITERIA DETERMINE THE CRITICAL KV CACHE?

Recent research has demonstrated only a small portion of critical KV cache entries do substantially contribute to the attention output (Zhang et al., 2024b; Liu et al., 2024b). This insight presents

promising opportunities to reduce inference costs by evicting a large number of non-critical KV cache entries (Li et al., 2024; Zhang et al., 2024a; Feng et al., 2024; Ge et al., 2024b; Adnan et al., 2024; Ge et al., 2024a). However, the key challenge lies in accurately identifying the critical KV cache entries. Ideally, from a high-level perspective, the set of critical KV cache entries should completely represent the entire cache, ensuring for given query state, the selected entries yield the same attention output as the full set of KV pairs. In practice, the number of selected critical cache entries will be constrained by a predefined budget, which is closely tied to the computational resources available in downstream deployments. Consequently, our goal shifts toward minimizing the output perturbation introduced by the replacement. So, the problem can be reformulated as follows.

**Definition 3.1** (Critical KV Cache Identification Problem)**.** Given a critical cache budget $b$, the task is to select $b$ critical KV cache entries $\langle \hat{K}, \hat{V} \rangle$ from a total of $n$ cache entries $\langle K, V \rangle$, with the goal of minimizing the perturbation in the attention output $o$. By using the $L_1$ distance $\mathcal{L}$ for quantification, the objective is formalized as: $\arg\min_{\langle \hat{K}, \hat{V} \rangle} \mathcal{L} = \|o - \hat{o}\|_1$ , where $\hat{o}$ represents the attention output produced by the selected $\langle \hat{K}, \hat{V} \rangle$.

### 3.3 ARE ATTENTION WEIGHTS SUFFICIENT FOR IDENTIFYING CRITICAL CACHE ENTRIES?

According to Definition 3.1, the goal of identifying critical KV cache entries is to minimize the perturbation $\mathcal{L} = \|o - \hat{o}\|_1$. To achieve this, we can employ an additive masking $\mathcal{M}$ to simulate the removal of non-critical cache entries' contributions to the final output $\hat{o}$, thereby altering $\hat{o}$.

$$\hat{o} = A'VW^O, \ A' = \text{softmax}\left(\mathcal{M} + qK^T/\sqrt{d}\right) \text{ where } \mathcal{M}_i = \begin{cases} -\infty & \text{if } K_i \text{ and } V_i \text{ are non-critical} \\ 0 & \text{otherwise.} \end{cases} \quad (3)$$

Thus, the perturbation $\mathcal{L}$ can be further expressed as: $\mathcal{L} = \|(A - A')VW^O\|_1$

**Theorem 3.2.** *By introducing a mask* $\mathcal{N} \in \mathbb{R}^n$ *applied through element-wise multiplication denoted by* $\odot$, *we can establish the relation between* $A'$ *and* $A$ *as follows:*

$$A' = \frac{\mathcal{N} \odot A}{\sum_{i=1}^n \mathcal{N}_i A_i} \quad \text{where } \mathcal{N}_i = \begin{cases} 0 & \text{if } K_i, V_i \text{ is non-critical} \\ 1 & \text{otherwise.} \end{cases} \quad \text{and} \sum_{i=1}^n \mathcal{N}_i = b \quad (4)$$

*Proof.* See Appendix K.1 for details. $\qquad\square$

Theorem 3.2 utilizes a multiplicative mask $\mathcal{N}$ to quantifies how their selection impacts the attention weights. However, directly minimizing $\mathcal{L}$ for critical cache selection is challenging due to complex matrix operations it requires. Thus we turn to establish an upper bound $\theta$, as shown in Theorem 3.3.

**Theorem 3.3.** *The output perturbation* $\mathcal{L}$ *can be bounded by* $\theta$:

$$\mathcal{L} \leq \theta = C - \left(2 - \frac{1}{\sum_{i=1}^n \mathcal{N}_i A_i}\right) \sum_{i=1}^n \mathcal{N}_i A_i \|\mathcal{V}_{i,:}\|_1, \quad (5)$$

*where* $C$ *denotes the* $\sum_{i=1}^n A_i \|\mathcal{V}_{i,:}\|_1$ *and* $\mathcal{V} \in \mathbb{R}^{n \times d} = VW^O$ *denotes all projected values states through parameter matrix* $W^O$.

*Proof.* See Appendix K.2 for details. $\qquad\square$

We can observe that $\theta$ encompasses not only the attention weights but also the projected value states. This highlights that prior selection methods relying solely on attention weights are suboptimal.

### 3.4 IDENTIFY CRITICAL CACHE ENTRIES BY CONSTRAINING WORST-CASE PERTURBATION.

Drawing on optimization strategies in machine learning, we propose lowering the upper bound of perturbation, effectively constraining the worst-case perturbation and thereby reducing actual perturbations for identifying critical cache entries. However, directly minimizing the upper bound $\theta$ remains non-trivial. To balance both the complexity and selection effectiveness, we introduce a two-stage greedy perturbation-constrained selection Algorithm 1, specifically designed to lower the perturbation upper bound for critical cache entry identification.

In this algorithm, the total budget $b$ is divided into two portions based on a hyperparameter $\alpha$. In the first stage, a fraction of the budget, $b' = b \times \alpha$, is allocated to prioritize KV cache entries with high attention weights. In the second stage, the remaining budget, $b'' = b - b'$, is used to consider both the

**Algorithm 1** Perturbation-Constrained Selection

**Input**: Budgets $b$, Query State $q$, Cache Entries $K, V$, Parameter Matrix $W^O$, Hyper Parameter $\alpha = 0.25$
**Output**: Critical Cache Entries $\hat{K}, \hat{V}$
 1: initialize empty cache $\hat{K}, \hat{V}$
 2: $A = \text{softmax}(qK^T); \boldsymbol{\mathcal{V}} = VW^O$
 3: $\boldsymbol{\mathcal{A}} = (A + \epsilon) \odot (L_1 \text{ norm of each rows in } \boldsymbol{\mathcal{V}})$
 4: $b' = b \times \alpha; b'' = b - b'$
 5: **for all** $K_i, V_i \in K, V$ that $A_i \in \text{Top}_k(A, b')$
 6:     add $K_i, V_i$ to $\hat{K}, \hat{V}$      **Stage 1**
 7:     remove $\boldsymbol{\mathcal{A}}_i, K_i, V_i$ from $\boldsymbol{\mathcal{A}}, K, V$
 8: **for all** $K_i, V_i \in K, V$ that $\boldsymbol{\mathcal{A}}_i \in \text{Top}_k(\boldsymbol{\mathcal{A}}, b'')$:
 9:     add $K_i, V_i$ to $\hat{K}, \hat{V}$      **Stage 2**
10: **return** Critical Cache Entries $\hat{K}, \hat{V}$

**Algorithm 2** Observation Win Based Eviction.

**Input**: All Query States $Q \in \mathbb{R}^{n \times d_h}$, KV Cache Entries $K, V \in \mathbb{R}^{n \times d_h}$, Window Size $n'$
**Output**: Critical Cache Entries $\hat{K}, \hat{V}$
 1: allocate budget $b$ across heads #AdaKV,HeadKV
 2: $\hat{Q} = Q[-n' :, :]$
 3: $A = \text{softmax}(\hat{Q}K^T); \bar{A} = A.\text{mean}(dim = 0)$
 4: $\bar{A} = \text{maxpooling}(\bar{A})$ # SnapKV
 5: **if** using regular selection **then**
 6:     select $b$ critical entries $\hat{K}, \hat{V}$ by $\text{Top}_k(\bar{A}', b)$
 7: **else if** using our selection **then**
 8:     select $b$ critical entries $\hat{K}, \hat{V}$ by Algorithm 1
 9: **end if**
10: **return** Critical Cache Entries $\hat{K}, \hat{V}$

norms of the projected value states and the attention weights [2]. This two-stage selection employs a Top-K operation to effectively constrain the worst-case perturbation. To substantiate the effectiveness of our proposed algorithm, we provide a theoretical analysis in the following section.

## 3.5 THEORETICAL ANALYSIS OF ALGORITHM 1

Our proposed algorithm consists of two stages, referred to as stage 1 and stage 2, which work collaboratively to select critical cache entries. Under the guarantee provided by Assumption 3.4, the selection in stage 1 ensures that stage 2 adheres to the constraints on perturbations, as formalized in Theorem 3.5. Let $\mathcal{N}'$ and $\mathcal{N}''$ represent the selections from the stage 1 and 2, respectively, satisfying: $\sum_{i=1}^{n} \mathcal{N}'_i = b'$ and $\sum_{i=1}^{n} \mathcal{N}''_i = b''$. Thus, the overall selection is $\mathcal{N} = \mathcal{N}' + \mathcal{N}''$.

**Assumption 3.4.** In the first stage, a portion of the overall budget $b' = b \times \alpha$ is sufficient to collect the cache entries corresponding to the highest attention weights, ensuring their cumulative attention weights $\sigma$ exceed half of the total, i.e., $\sigma = \sum_{i=1}^{n} \mathcal{N}'_i A_i = \sum \text{Top}_k(A, b') > 0.5$.

In this paper, we set $\alpha$ in Assumption 3.4 to a fixed value 0.5 based on two key considerations. First, as verified in Appendix A, allocating 50% of the total budget is sufficient to capture enough attention weight in over 99% of attention heads, thereby satisfying Assumption 3.4 across various settings. This is attributed to the power-law distribution of attention weights (Zhang et al., 2024b), where a small fraction of cache entries accounts for the majority. Second, this choice is both robust and easy to apply across different cache budgets and models. While using different $\alpha$ values for specific models, budgets, or attention heads could yield finer optimization, it would also introduce significant search overhead and complicate deployment. Thus, we defer such granular adjustments to future work. Subsequent experiments and visual analyses further confirm that setting $\alpha$ to 0.5 is a simple yet effective choice. [3]

**Theorem 3.5.** *Given the stage 1 selection $\mathcal{N}'_i$, the objective $\mathcal{N}''_i$ of stage 2 is to minimize an upper bound $\hat{\theta}$ of the output perturbation $\mathcal{L}$, using the remaining budget $b'' = b - b'$.*

$$\underset{\mathcal{N}''_i}{\arg\min} \, \hat{\theta} \text{ where } \hat{\theta} = C' - \left(2 - \frac{1}{\sigma}\right) \sum_{i=1}^{n} \mathcal{N}''_i A_i \|\boldsymbol{\mathcal{V}}_{i,:}\|_1$$

$$\text{subject to } \sum_{i=1}^{n} \mathcal{N}''_i = b'', C' = C - \left(2 - \frac{1}{\sigma}\right) \sum_{i=1}^{n} \mathcal{N}'_i A_i \|\boldsymbol{\mathcal{V}}_{i,:}\|_1. \tag{6}$$

*Proof.* See Appendix K.3 for details. □

Theorem 3.5 demonstrates that our second stage selection directly minimizes an upper bound of output perturbation for identifying critical cache entries. Unlike traditional strategies that rely solely on high attention weights for entry selection, the second stage of our algorithm jointly leverages both the attention weights and the value states projected through the parameter matrix, to directly constrain the worst-case output perturbation.

---

[2] A small $\epsilon$ (1E-4) is added to mitigate information loss from sparse attention weights during multiplication.
[3] As shown in Section 4.5, our algorithm's performance is robust to the choice of hyperparameter $\alpha$.

### 3.6 INTEGRATING INTO SOTA CACHE EVICTION METHODS

We showcase the effectiveness of our algorithm by integrating it into existing cache eviction methods that rely on accumulated attention weights for selecting critical entries. Current SOTA plug-and-play cache eviction workflow is established by SnapKV (Li et al., 2024), which introduces an observation window mechanism to stably accumulate attention weights and employs the max pooling operations to avoid missing key information. Subsequent research (Zhang et al., 2024a; Feng et al., 2024) highlights the uneven distribution of critical cache entries across different heads, prompting the development of budget allocation strategies. For example, AdaKV (Feng et al., 2024) improves upon SnapKV by dynamically detecting variations in critical KV cache entries at runtime, enabling flexible budget scheduling and enhancing output quality. Other methods, such as HeadKV (Fu et al., 2024), further refine budget scheduling, albeit at the cost of offline training. Despite their differences in budget allocation, these SOTA methods—SnapKV, AdaKV, and HeadKV—all use the same underlying mechanism for KV cache selection. Consequently, they can be unified under the framework of Algorithm 2, which consists of two main components: budget allocation across heads (line 1) and an observation window with a pooling mechanism for attention weight accumulation (lines 2–5). Our algorithm integrates seamlessly into this framework by replacing the original selection process, which is based solely on attention weights (lines 5–9).

## 4 EXPERIMENTS

### 4.1 SETTINGS

**Models.** We select three advanced open-source LLMs for evaluation: Llama-3.1-8B-Instruct (Llama-3.1-8B) (Dubey et al., 2024), Mistral-7B-Instruct-v0.3 (Mistral-7B) (Jiang et al., 2023), and Qwen-2.5-32B (Team, 2024). These models span different model families and parameter scales, demonstrating the broad applicability of our algorithm.

**Compression scenario.** Following (Feng et al., 2024; NVIDIA, 2024), the context is compressed independently before question is introduced. This setting better simulates practical scenarios (e.g., multi-turn QA or prefixed contexts) where multiple questions often pertain to the same context, or the question is unavailable during context compression. It therefore provides a more realistic evaluation of cache eviction methods. For the simple compression setting, where the context and question are compressed together, please refer to Appendix F

**Baselines.** We integrated our algorithm with three cache eviction methods—SnapKV (Li et al., 2024), AdaKV (Feng et al., 2024) and HeadKV (Fu et al., 2024). These respectively represent the SOTA cache eviction in non–budget-allocation, adaptive budget allocation, and offline budget allocation. By comparing the quality before and after integration, we demonstrate the improvements our algorithm brings to these methods. We set $\alpha = 0.5$ in Algorithm 1 for all experiments; see Section 4.5 for robustness analysis. SnapKV and AdaKV used their original settings: max-pooling kernel size 7 and observation window 32 (Feng et al., 2024). All methods were implemented with FlashAttention-2 for faster inference. We also include the performance of H2O (Zhang et al., 2024b) for reference, a foundational work in this area. Since it requires global attention weights—unsupported by FlashAttention-2—it triggers OOM. Following (Xiao et al., 2024c), we simulate H2O by observing the last 256 tokens' attention weights.

### 4.2 RULER BENCHMARK

The Ruler benchmark (Hsieh et al., 2024) comprises 13 synthetic tasks for evaluating long-context capabilities—a challenging testbed for cache eviction. It includes two Word Extraction variants (CWE and FWE)—eight Needle-In-A-Haystack variations (NIAH), as well as Question Answering(QA) and Variable Tracking (VT), with each scored out of 100. To match the Mistral model's 32K context window and control cost, we set the ruler length to 32K and sampled 100 instances per task.

Table 1 reports task-wise scores at a 40% cache size. SnapKV, AdaKV, and HeadKV all degrade under cache eviction, but each sees substantial gains when augmented with our algorithm. For instance, on Qwen2.5-32B, our algorithm yields consistent gains across all 13 tasks for all three eviction method. Quantitatively, our algorithm increases the average score of SnapKV from 63.86 to 81.09 and AdaKV from 71.09 to 83.87. When applied to HeadKV, the average score climbs from

Table 1: Detail Results on Ruler Benchmark with 40% Cache Size

| | | CWE | FWE | NIAH Multikey1 | NIAH Multikey2 | NIAH Multikey3 | NIAH Multiquery | NIAH Multivalue | NIAH Single1 | NIAH Single2 | NIAH Single3 | QA1 | QA2 | VT | Avg. Score ↓loss |
|---|---|---|---|---|---|---|---|---|---|---|---|---|---|---|---|
| **Llama-3.1-8B** | Full Cache | 44.90 | 95.33 | 100.00 | 100.00 | 100.00 | 98.25 | 99.75 | 100.00 | 100.00 | 100.00 | 84.00 | 62.00 | 99.40 | 91.05 ↓ 0.0% |
| | SnapKV | 20.30 | 90.00 | 93.00 | 36.00 | 31.00 | 92.75 | 87.50 | 100.00 | 97.00 | 46.00 | 41.00 | 52.00 | 96.60 | 67.93 ↓ 25.4% |
| | *w/ ours* | **33.90** | **92.67** | **100.00** | **50.00** | **32.00** | **99.00** | **99.00** | 100.00 | 100.00 | **80.00** | **63.00** | **53.00** | **97.00** | **76.89 ↓ 15.6%** |
| | AdaKV | 26.60 | 92.00 | 98.00 | 66.00 | **71.00** | 97.50 | 98.25 | 99.00 | 100.00 | 75.00 | 43.00 | 54.00 | 98.60 | 78.38 ↓ 13.9% |
| | *w/ ours* | **52.40** | **94.67** | **100.00** | **93.00** | 65.00 | **99.25** | **98.75** | 100.00 | 100.00 | **97.00** | **64.00** | **58.00** | **99.60** | **86.28 ↓ 5.2%** |
| | HeadKV | 17.10 | 92.67 | 99.00 | 66.00 | 75.00 | 97.00 | 93.00 | **100.00** | 98.00 | 94.00 | 56.00 | 54.00 | 98.00 | 79.98 ↓ 12.2% |
| | *w/ ours* | **56.50** | **93.33** | **100.00** | **93.00** | **90.00** | **99.50** | **99.00** | 100.00 | 100.00 | **99.00** | **72.00** | **59.00** | **99.40** | **89.29 ↓ 1.9%** |
| **Mistral-7B** | Full Cache | 30.00 | 95.67 | 94.00 | 69.00 | 31.00 | 94.25 | 95.00 | 99.00 | 100.00 | 100.00 | 60.00 | 59.00 | 90.60 | 78.27 ↓ 0.0% |
| | SnapKV | 32.20 | 91.67 | 16.00 | **11.00** | 4.00 | 11.25 | 8.00 | 65.00 | 17.00 | **2.00** | 34.00 | 51.00 | 74.80 | 32.15 ↓ 58.9% |
| | *w/ ours* | **48.50** | **94.67** | **31.00** | 7.00 | **5.00** | **26.00** | **24.25** | **74.00** | **48.00** | 2.00 | **45.00** | **53.00** | **81.80** | **41.56 ↓ 46.9%** |
| | AdaKV | 25.30 | 92.33 | 25.00 | 14.00 | **8.00** | 20.00 | 14.25 | 44.00 | 34.00 | 6.00 | 44.00 | 55.00 | 71.60 | 34.88 ↓ 55.4% |
| | *w/ ours* | **40.50** | **95.33** | **89.00** | **27.00** | 8.00 | **85.00** | **95.00** | **99.00** | **100.00** | **54.00** | **60.00** | **57.00** | **89.40** | **69.17 ↓ 11.6%** |
| | HeadKV | 30.60 | 92.67 | 24.00 | 27.00 | **17.00** | 16.25 | 16.50 | 70.00 | 36.00 | 4.00 | 45.00 | 53.00 | **82.60** | 39.59 ↓ 49.4% |
| | *w/ ours* | **49.50** | **95.00** | **53.00** | **31.00** | 17.00 | **60.50** | **62.25** | **84.00** | **82.00** | **25.00** | **53.00** | **55.00** | 81.40 | **57.59 ↓ 26.4%** |
| **Qwen2.5-32B** | Full Cache | 90.60 | 96.00 | 99.00 | 90.00 | 90.00 | 100.00 | 99.25 | 100.00 | 100.00 | 100.00 | 82.00 | 74.00 | 100.00 | 93.91 ↓ 0.0% |
| | SnapKV | 87.90 | 92.00 | 63.00 | 23.00 | 9.00 | 71.75 | 61.75 | 100.00 | 94.00 | 14.00 | 50.00 | 64.00 | 99.80 | 63.86 ↓ 32.0% |
| | *w/ ours* | **88.60** | **93.33** | **98.00** | **35.00** | **16.00** | **99.75** | **99.50** | 100.00 | 100.00 | **88.00** | **67.00** | **69.00** | **100.00** | **81.09 ↓ 13.7%** |
| | AdaKV | 88.90 | 93.33 | 81.00 | 34.00 | 21.00 | 87.50 | 83.50 | 100.00 | 97.00 | 19.00 | 55.00 | 64.00 | 100.00 | 71.09 ↓ 24.3% |
| | *w/ ours* | **89.50** | **94.33** | **98.00** | **54.00** | **34.00** | **100.00** | **99.50** | 100.00 | 100.00 | **91.00** | **64.00** | **66.00** | 100.00 | **83.87 ↓ 10.7%** |
| | HeadKV | 89.30 | 94.67 | 92.00 | 46.00 | 45.00 | 97.25 | 96.25 | **100.00** | **100.00** | 59.00 | 68.00 | 66.00 | **100.00** | 81.04 ↓ 13.7% |
| | *w/ ours* | **89.90** | **95.33** | **99.00** | **85.00** | **71.00** | **100.00** | **98.75** | 100.00 | 100.00 | **72.00** | **69.00** | **69.00** | 100.00 | **90.69 ↓ 3.4%** |

Legend: AdaKV · AdaKV w/ ours · HeadKV · HeadKV w/ ours · SnapKV · SnapKV w/ ours · H2O

Figure 2: Performance on Ruler Tasks with Varying Cache Sizes

81.04 to 90.69, mitigating the loss relative to the full cache from 13.7% to just 3.4%. Similar trends hold for the Llama and Mistral models, with an average improvement of 14.6 points across all cases.

Figure 2 further offers a comprehensive view across different cache sizes for both the Llama and Mistral models. Results for Qwen2.5-32B are omitted due to the prohibitive cost of evaluating a 32B-scale model across multiple cache sizes. More results for the Mistral are provided in Appendix G. On both the Llama and Mistral models, our method consistently and significantly improves all base methods. For instance, with Llama model at a 60% cache size, AdaKV achieves an average score of 87.92. When enhanced with our algorithm, this score increases to 90.94—almost matching the full-cache performance of 91.04. The benefits of our algorithm are even more pronounced at small cache sizes. at 40% and 20%, our algorithm increases AdaKV's average scores from 78.38 to 86.28 and from 57.48 to 68.94, respectively. Similarly, on Mistral, our algorithm boosts AdaKV from 48.80 to 75.85 at 60% size, and from 34.88 to 69.17 at 40% size. These results highlight the effectiveness of our algorithm as a universal enhancement to a wide range of existing KV cache eviction methods.

## 4.3 LONGBENCH EVALUATION

We also incorperate the real-world benchmark LongBench, consisting of 16 datasets across six task domains: single-document QA, multi-document QA , summarization, few-shot learning, synthetic,

Table 2: Task Domain Scores on LongBench.

| Domain | Full Cache | SnapKV 20%b | | SnapKV 40%b | | AdaKV 20%b | | AdaKV 40%b | | HeadKV 20%b | | HeadKV 40%b | |
|---|---|---|---|---|---|---|---|---|---|---|---|---|---|
| | | base | w/ ours | base | w/ ours | base | w/ ours | base | w/ ours | base | w/ ours | base | w/ ours |
| **Llama-3.1-8B** | | | | | | | | | | | | | |
| SingleDoc. QA | 43.10 | 28.78 | **30.43** | 35.27 | **38.27** | 31.39 | **32.74** | 36.63 | **39.24** | 31.53 | **33.60** | 39.96 | **40.61** |
| MultiDoc. QA | 46.49 | 33.51 | **35.87** | 40.50 | **43.17** | 34.90 | **35.31** | 41.36 | **45.11** | 34.97 | **36.33** | 43.10 | **44.33** |
| Summarization | 28.97 | 23.82 | **24.64** | 26.11 | **27.15** | 24.29 | **24.98** | 26.66 | **27.31** | 24.49 | **25.28** | 27.06 | **27.24** |
| Fewshot | 69.45 | 61.95 | **63.04** | 65.10 | **66.87** | 63.70 | **64.74** | 66.43 | **68.19** | 62.79 | **65.41** | 65.64 | **68.06** |
| Synthetic | 53.73 | 48.19 | **52.16** | 53.17 | **54.22** | 50.39 | **52.30** | 53.00 | **53.60** | 49.32 | **52.18** | 52.89 | **54.04** |
| Code | 57.86 | 60.05 | **60.75** | 60.49 | **60.91** | 61.14 | **61.16** | 60.30 | **60.60** | 61.14 | 58.85 | **61.34** | 57.89 |
| Avg. Score | 49.20 | 41.29 | **42.99** | 45.52 | **47.29** | 42.87 | **43.77** | 46.24 | **48.00** | 42.64 | **43.99** | 47.23 | **47.79** |
| Avg. Loss ↓ | 0.00 % | 16.1 % | **12.6** % | 7.5 % | **3.9** % | 12.9 % | **11.0** % | 6.0 % | **2.4** % | 13.3 % | **10.6** % | 4.0 % | **2.9** % |
| **Mistral-7B** | | | | | | | | | | | | | |
| Single-Doc. QA | 38.37 | 25.13 | **28.24** | 31.64 | **34.33** | 27.25 | **29.06** | 33.42 | **36.06** | 27.94 | **30.95** | 33.90 | **36.21** |
| Multi-Doc. QA | 39.40 | 29.64 | **32.07** | 33.57 | **36.61** | 31.71 | **33.04** | 35.97 | **38.00** | 31.58 | **34.47** | 35.28 | **37.66** |
| Summarization | 28.76 | 24.18 | **24.61** | 26.24 | **26.94** | 24.18 | **24.83** | 26.24 | **27.06** | 24.49 | **25.63** | 26.76 | **27.72** |
| Few-shot | 70.33 | 63.89 | **65.54** | 67.95 | **68.57** | 66.00 | **68.08** | 68.67 | **69.79** | 64.73 | **67.58** | 67.46 | **69.78** |
| Synthetic | 52.50 | 45.25 | **46.53** | 49.75 | **51.05** | 48.00 | **49.25** | 50.75 | **50.84** | 47.79 | **48.02** | 50.50 | 50.20 |
| Code | 61.25 | 61.40 | **61.94** | 63.41 | 62.06 | 62.55 | **62.56** | 63.35 | 62.68 | **61.96** | 61.76 | 63.16 | 61.56 |
| Avg. Score | 47.38 | 40.11 | **41.77** | 44.03 | **45.35** | 41.78 | **42.85** | 45.07 | **46.23** | 41.61 | **43.46** | 44.84 | **46.10** |
| Avg. Loss ↓ | 0.0 % | 15.3 % | **11.8** % | 7.1 % | **4.3** % | 11.8 % | **9.6** % | 4.9 % | **2.4** % | 12.2 % | **8.3** % | 5.4 % | **2.7** % |
| **Qwen2.5-32B** | | | | | | | | | | | | | |
| Single-Doc. QA | 43.23 | 26.65 | **28.12** | 32.22 | **37.18** | 26.62 | **29.00** | 32.37 | **35.70** | 29.9 | **32.07** | 38.74 | **40.49** |
| Multi-Doc. QA | 54.03 | 42.00 | **46.88** | 50.90 | **54.55** | 42.36 | **47.40** | 52.75 | **53.75** | 46.35 | **50.60** | 55.13 | **55.33** |
| Summarization | 27.40 | 23.22 | **24.23** | 25.19 | **26.03** | 23.21 | **24.06** | 24.95 | **25.97** | 23.88 | **24.34** | 26.10 | **26.36** |
| Few-shot | 68.97 | 65.43 | **66.86** | 67.43 | **67.80** | 66.56 | **67.44** | 68.18 | **68.65** | 65.96 | **68.53** | 68.37 | **69.27** |
| Synthetic | 56.25 | 44.21 | **53.00** | 55.25 | **55.75** | 46.63 | **52.75** | 54.63 | **55.25** | 50.88 | **53.88** | 55.09 | 55.04 |
| Code | 41.93 | **45.04** | 44.72 | **44.89** | 43.75 | **46.30** | 45.43 | **46.21** | 44.88 | 44.83 | **46.27** | 44.88 | **46.54** |
| Avg. Score | 48.58 | 40.65 | **43.36** | 45.47 | **47.23** | 41.38 | **43.75** | 46.03 | **47.03** | 43.11 | **45.43** | 47.81 | **48.59** |
| Avg. Loss ↓ | 0.0 % | 16.3 % | **10.8** % | 6.4 % | **2.8** % | 14.8 % | **9.9** % | 5.3 % | **3.2** % | 11.3 % | **6.5** % | 1.6 % | **0.0** % |

and code. For each dataset, we use the recommended evaluation metrics and report the average score within each task domain. Detailed information can be found in Appendix H.

As shown in Table 2, our algorithm achieves improvements across most evaluation cases. In the five widely used long-dependency task domains (single-document QA, multi-document QA , summarization, few-shot learning, synthetic), cache eviction degrades the performance by disrupting historical information. Our algorithm markedly mitigates this loss: across 90 test cases—covering five long-dependency domains, three models (Llama-3.1-8B, Mistral-7B, Qwen2.5-32B), and three compression methods (SnapKV, AdaKV, HeadKV) at two cache sizes—we observe improvements in 88 cases. This 97.8% success rate highlights the breadth and robustness of our method. Numerically, the effect is clear: for instance in Multi-Doc QA with Llama-3.1-8B, applying AdaKV at 40% cache reduces the score from 46.49 to 41.36; adding our algorithm raises it to 45.11. On Mistral-7B, compression lowers the score from 39.40 to 35.97, while our algorithm restores it to 38.00. By contrast, the code domain is naturally insensitive to cache eviction. At a 20% cache size, performance can even surpass that of the full cache—a phenomenon reported in prior work Feng et al. (2024); Li et al. (2024); Zhang et al. (2024a). This arises because code-related tend to rely less on long-range dependencies; compressing the KV cache can paradoxically improve accuracy by filtering out historical context. Consequently, the code domain is generally not considered a suitable indicator.

With respect to the average performance loss compared to full cache case, our method substantially reduces this degradation from previous cache eviction methods. For instance, under a 40% cache size on the Llama model, integrating our algorithm with AdaKV reduces the average loss from 6.0% to 2.4%. Similar gains are observed for the Mistral and Qwen models, where our algorithm decreases the losses from 4.9% to 2.4% and from 5.3% to 3.2%, respectively. These results affirm that our algorithm provides a robust and general solution for mitigating the quality loss caused by KV cache eviction across diverse real-world applications.

## 4.4 MULTI-TURN QA EVALUATION

To further evaluate the effectiveness of our proposed method in processing long-horizon information and handling future-unknown questions, we conducted experiments on SCBench (Li et al., 2025) designed for multi-turn dialogue and long-context compression.

**Settings.** SCBench comprises 4,853 QA turns with an average context length of 227K tokens. This scale significantly exceeds the effective context window of most open-source LLMs, including the Llama-3.1-8B used in our experiments (128K context window). To accommodate these constraints while reserving sufficient capacity for generation in multi-turn scenarios, we selected three repre-

sentative tasks across different retrieval categories: *Retr.KV* (String Retrieval), *EN.QA* (Semantic Retrieval), and *Math.Find* (Global Information). For these tasks, we truncated the input contexts to 100K tokens to prevent context overflow during subsequent dialogue turns.

**Results.** We compared the performance of the vanilla AdaKV baseline against AdaKV enhanced with our method across three cache budget settings: 80%, 60%, and 40%. The results are summarized in Table 3. Our method demonstrates consistent improvements over the baseline across all tasks. Notably, the performance gain becomes more pronounced as the cache budget decreases. For instance, in the *EN.QA* task at a 40% compression rate, our method achieves a score of 22.14 compared to the baseline's 15.71, representing a substantial improvement. These results indicate that our method effectively enhances the ability to preserve critical information.

Table 3: Performance comparison on SCBench multi-turn QA tasks.

| Task (Turns) | Full Cache | 80% Cache Budget | | 60% Cache Budget | | 40% Cache Budget | |
|---|---|---|---|---|---|---|---|
| | | AdaKV | + Ours | AdaKV | + Ours | AdaKV | + Ours |
| Retr.KV (5 turns) | 52.20 | 52.00 | **52.60** | 41.20 | **42.00** | 19.40 | **19.80** |
| Math.Find (6 turns) | 11.67 | 11.33 | **13.67** | 11.67 | **14.17** | 11.50 | **16.83** |
| EN.QA (7 turns) | 22.86 | 22.14 | **24.29** | 17.86 | **21.43** | 15.71 | **22.14** |

## 4.5 ANALYSIS OF $\alpha$

As demonstrated previously, we introduce the hyperparameter $\alpha$ as a safeguard to satisfy Assumption 3.4. To investigate the sensitivity and necessity of this parameter, we evaluate the performance of our method (integrated with AdaKV) on LongBench across a range of $\alpha$ values $\{0.0, 0.3, 0.5, 0.7\}$ with a 20% cache budget. [4] The detailed results are presented in Table 5.

Empirically, we observe that a simple choice of $\alpha = 0.5$ is sufficient to ensure robust performance across different models over base method AdaKV. For the Llama-3.1-8B model, the performance remains relatively stable across different $\alpha$ values, indicating the robustness of $\alpha$. However, the results on Mistral-7B-v0.3 highlight the critical necessity of this safeguard. Removing $\alpha$ entirely (i.e., setting $\alpha = 0$) leads to a violation of the assumption for certain attention heads, resulting in significant performance degradation. Specifically, as shown in Table 4, setting $\alpha = 0$ on Mistral causes the score to 31.94, lagging behind the default setting ($\alpha = 0.5$, 42.85) by over 10 points.

Table 4: Sensitivity analysis of the hyperparameter $\alpha$ on LongBench with 20% cache budget.

| Model / Setting | Multi-Doc | Single-Doc | Sum. | Few-Shot | Synthetic | Code | Avg. |
|---|---|---|---|---|---|---|---|
| *Llama-3.1-8B (20% Cache)* | | | | | | | |
| AdaKV | 34.03 | 29.54 | 23.74 | 63.56 | 43.90 | 60.94 | 41.39 |
| AdaKV w/ ours ($\alpha = 0.0$) | **38.14** | 33.56 | **25.14** | 64.25 | 51.73 | **61.41** | **44.35** |
| AdaKV w/ ours ($\alpha = 0.3$) | 36.73 | **33.80** | 24.94 | 63.29 | 50.21 | 61.03 | 43.67 |
| AdaKV w/ ours ($\alpha = 0.5$) | 35.31 | 32.74 | 24.98 | **64.74** | **52.30** | 61.16 | 43.77 |
| AdaKV w/ ours ($\alpha = 0.7$) | 36.95 | 32.20 | 24.70 | 63.36 | 49.14 | 61.38 | 43.29 |
| *Mistral-7B-v0.3 (20% Cache)* | | | | | | | |
| AdaKV | 31.30 | 27.15 | 23.81 | 65.05 | 46.25 | 62.27 | 41.18 |
| AdaKV w/ ours ($\alpha = 0.0$) | 26.12 | 22.49 | 23.42 | 39.44 | 31.29 | 56.97 | 31.94 |
| AdaKV w/ ours ($\alpha = 0.3$) | 31.69 | **30.69** | **24.87** | **67.30** | 46.84 | 61.67 | 42.54 |
| AdaKV w/ ours ($\alpha = 0.5$) | **33.04** | 29.06 | 24.83 | 67.08 | **49.25** | **62.56** | **42.85** |
| AdaKV w/ ours ($\alpha = 0.7$) | 32.60 | 30.11 | 24.78 | 66.90 | 49.00 | 61.82 | 42.80 |

## 4.6 EFFICIENCY EVALUATION

Cache eviction methods are typically applied after the prefill phase to significantly save cache memory footprint and accelerate decoding. We evaluate efficiency using time-to-first-token (TTFT) for the prefill phase (including eviction process) and single-step latency for decoding, measured on a single

---

[4]The results of on a small 10% budget are shown in Appendix C

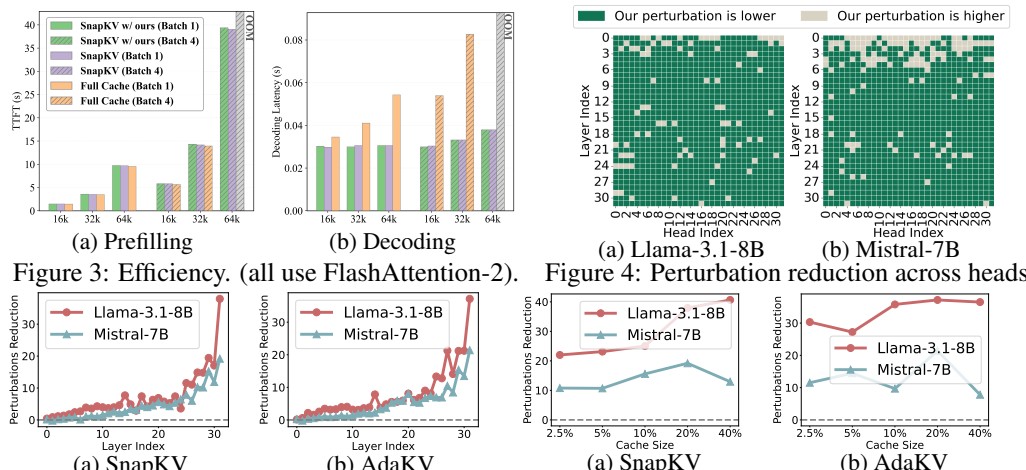

Figure 3: Efficiency. (all use FlashAttention-2).  Figure 4: Perturbation reduction across heads.

Figure 5: Perturbation reduction across layers.  Figure 6: Perturbation reduction across budgets.

80GB A100 GPU with Llama-3.1-8B and a 40% cache size. The additional overhead introduced by our method is a minor increase in TTFT from the perturbation constraints algorithm, primarily due to the computation of $|VW^O|$. This operation is linear in complexity and has negligible impact. As shown in Figure 3a, at a 32K context length, TTFT increases by only 0.06s for batch size 1 ($3.54 \rightarrow 3.60$) and 0.16s for batch size 4 ($14.20 \rightarrow 14.36$, or 0.04s per request). For decoding, all cache eviction methods demonstrate same efficiency and outperform the full cache baseline. For batch size 4 and 32K context, SnapKV (with or without our algorithm) achieves 0.0332s, representing a $2.49\times$ speedup over the full cache time of 0.0828s. Thus, our algorithm substantially improves the quality of existing cache eviction methods while maintaining almost identical computational efficiency.

### 4.7 ANALYSIS OF PRACTICAL OUTPUT PERTURBATION

We further investigate whether our algorithm of constraining the theoretical perturbation upper bound effectively reduces the practical output perturbation. We visualize the perturbations on attention output of the first decoding token, using 200 samples from the MultiNews summarization dataset with the KV cache compressed to 20% size. **a. Head-wise Analysis:** Our algorithm significantly reduces head-wise average output perturbation across all samples in the Llama model, achieving lower perturbations in 92% and 86% of attention heads for Llama-3.1-8B and Mistral-7B, respectively (Figure 4). **b. Layer-wise Analysis:** Figure 5 shows how our algorithm progressively reduces perturbation across layers, leading to substantial decreases in the final layer, which directly impacts the generated token vocabulary distribution. **c. Budget-wise Analysis:** Figure 6 illustrates that our method effectively lowers output perturbation across different cache sizes from 2.5% to 40%, underscoring its robustness of varying budget constrains. These analyses demonstrate that our method robustly reduces practical output perturbation by theoretically constraining worst-case perturbation by Algorithm 1. This results in the post-eviction output hidden states that are more consistent with those from the full KV cache, thereby enhancing generation consistency and reducing quality loss.

## 5 CONCLUSION

In this paper, we pinpoint a key limitation in current cache eviction methods: the reliance on intuitive heuristics of using attention weights to select critical cache entries. For the first time, we formalize the problem of critical cache entry selection from the perspective of output perturbation and provide a theoretical analysis. Furthermore, we propose a novel algorithm based on constraining output perturbation in the worst-case for critical cache selection, which is then integrated into existing SOTA cache eviction methods. Comprehensive evaluations using 29 datasets from Ruler and Longbench demonstrate that our algorithm improves the performance of existing cache eviction methods. Further empirical analysis also confirms and explains this benefit from the perspective of practical output perturbation: our algorithm consistently yields lower perturbation compared to previous methods that rely solely on attention weights in various settings. Our work offers a new perspective for advancing cache eviction area, highlighting its significant benefits and future potential.

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

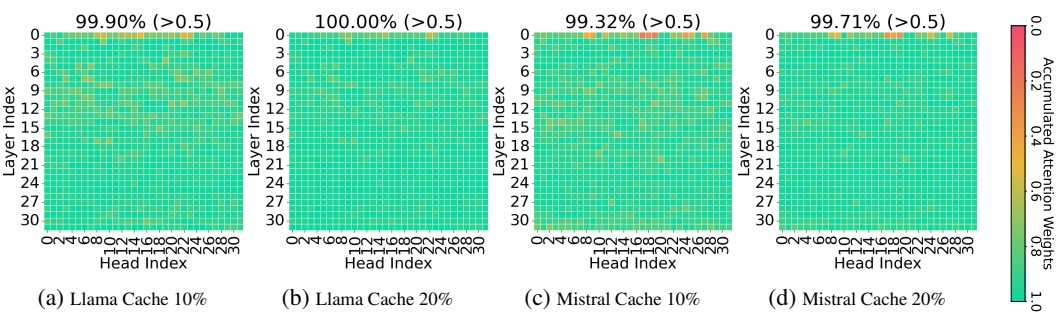

(a) Llama Cache 10%  (b) Llama Cache 20%  (c) Mistral Cache 10%  (d) Mistral Cache 20%

Figure 7: Assumption 3.4 validates in over 99% of heads across various cache sizes.

# A    ANALYSIS OF $\alpha$ IN ASSUMPTION 3.4

We ensure the reliability of Assumption 3.4 by analyzing the cumulative attention weights of critical KV Cache entries $\sum_{i=1}^{n} \mathcal{N}iAi$ in individual heads. As shown in Figure 7, across different models and cache sizes, almost all attention heads can accumulate over half of the attention weights as said in Assumption 3.4. The only exceptions are a few attention heads in the first layer. This is primarily due to the low sparsity of attention weights in certain heads of the first layer, a phenomenon that has been noted in many related studies (Tang et al., 2024a; Zhang et al., 2024b;a). However, this effect is negligible, as these heads constitute less than 1% of the total and their minor negative impact is far outweighed by the substantial gains from the compliant heads. A potential solution is to set the algorithm's threshold $\alpha$ based on the head-wise characteristics to achieve greater benefits. However, considering the additional complexity that might introduce, we retain a fixed $\alpha = 0.5$ for its simplicity and strong empirical performance, leaving fine-grained tuning for future work.

Table 5: Impact of safeguard $\alpha$ on algorithm performance based on Ada-KV.

| Model | Ada-KV | $\alpha = 0$ | $\alpha = 0.3$ | $\alpha = 0.4$ | $\alpha = 0.5$ | $\alpha = 0.6$ | $\alpha = 0.7$ |
|---|---|---|---|---|---|---|---|
| Llama-3.1-8B | 42.87 | 44.35 | 43.67 | 43.86 | 43.88 | 43.31 | 43.29 |
| Mistral-7B-v0.3 | 41.78 | 31.94 | 42.54 | 42.76 | 42.88 | 42.98 | 42.80 |

# B    COMPARISON WITH RECENT BASELINES

To further evaluate the effectiveness and robustness of our proposed method, we incorporate comparisons with additional two baselines: **PyramidKV** Zhang et al. (2024a) and **DuoAttention** Xiao et al. (2024b). We conduct experiments on the LongBench benchmark across 16 datasets, following the same setting as our main evaluation. The average scores are reported in Table 6.

As shown in the table, our method (applied to AdaKV and HeadKV) consistently outperforms PyramidKV across all settings. More importantly, while DuoAttention demonstrates competitive performance in high-budget scenarios, it lacks stability under stricter compression ratios. For instance, on Llama-3.1-8B with 40% cache size, DuoAttention achieves a score of 48.17, which is comparable to our method (AdaKV w/ ours: 48.00). However, its performance degrades significantly when the cache budget is limited or when transferred to other LLMs. Numerically, the contrast is sharp on the Mistral-7B model at a 20% cache size: DuoAttention drops to a score of 31.13, lagging behind our method (HeadKV w/ ours: 43.46) by over 12 points. Similarly, compared to AdaKV w/ ours (42.85), DuoAttention underperforms by approximately 11.7 points. This distinct performance gap highlights the robustness and effectiveness of our algorithm.

# C    EVALUATION ON EXTREME COMPRESSION (10% BUDGET)

In this section, we extend our evaluation to a more aggressive compression scenario with a 10% cache budget. This setting poses a significant challenge as it requires the model to discard 90% of the historical context while maintaining reasoning capabilities. The detailed results across different $\alpha$ values are reported in Table 7. Consistent with our observations at the 20% budget, the Llama-3.1-8B

Table 6: Performance comparison with recent baselines on LongBench (Average score of 16 datasets). **Bold** indicates the best performance.

| Method | Llama 20% Cache | Llama 40% Cache | Mistral 20% Cache | Mistral 40% Cache |
|---|---|---|---|---|
| PyramidKV | 40.22 | 44.20 | 39.77 | 43.30 |
| DuoAttention | 39.52 | **48.17** | 31.13 | 43.74 |
| AdaKV w/ ours | 43.77 | 48.00 | 42.85 | **46.23** |
| HeadKV w/ ours | **43.99** | 47.79 | **43.46** | 46.10 |

model remains relatively insensitive to the choice of $\alpha$, maintaining a stable performance around 38.5 points. This suggests that for robust models, even extreme compression does not trigger the failure modes that our method is designed to prevent. However, on the Mistral-7B-v0.3 model, the benefit of our proposed safeguard becomes evident. Without the safeguard (i.e., $\alpha = 0$), the performance drops to 35.94. By setting $\alpha = 0.5$, our method effectively recovers the performance to 37.94, yielding a 2.0-point improvement. This demonstrates that even under extreme memory constraints, our algorithm provides a crucial safety net for ensuring the quality of KV cache eviction on sensitive architectures.

Table 7: Performance on LongBench with an extreme **10% cache budget**.

| Model / Setting | Multi-Doc | Single-Doc | Sum. | Few-Shot | Synthetic | Code | Avg. |
|---|---|---|---|---|---|---|---|
| *Llama-3.1-8B (10% Cache)* | | | | | | | |
| AdaKV | 26.44 | 24.75 | 21.75 | 60.74 | 27.91 | **60.69** | 36.14 |
| AdaKV w/ ours ($\alpha = 0.0$) | 29.95 | 26.80 | 22.74 | 61.07 | **37.19** | 60.56 | **38.57** |
| AdaKV w/ ours ($\alpha = 0.3$) | 29.88 | 26.71 | 22.67 | **61.39** | 35.95 | 60.46 | 38.43 |
| AdaKV w/ ours ($\alpha = 0.5$) | **30.21** | **27.05** | **22.80** | 61.35 | 35.44 | 60.44 | 38.50 |
| AdaKV w/ ours ($\alpha = 0.7$) | 29.31 | 26.83 | 22.54 | 61.17 | 31.79 | 60.48 | 37.75 |
| *Mistral-7B-v0.3 (10% Cache)* | | | | | | | |
| AdaKV | 27.56 | 22.78 | 22.14 | 62.24 | **34.00** | **61.77** | 37.23 |
| AdaKV w/ ours ($\alpha = 0.0$) | 23.31 | 23.29 | 22.27 | **63.61** | 28.34 | 60.42 | 35.94 |
| AdaKV w/ ours ($\alpha = 0.3$) | 25.08 | 24.49 | **22.85** | 63.50 | 33.15 | 60.86 | 37.24 |
| AdaKV w/ ours ($\alpha = 0.5$) | **28.79** | **25.33** | 22.72 | 63.35 | 32.64 | 60.57 | **37.94** |
| AdaKV w/ ours ($\alpha = 0.7$) | 28.70 | 24.65 | 22.60 | 62.38 | 33.00 | 60.51 | 37.62 |

## D    EFFICIENCY WITH CHUNKED PREFILLING

In practical deployment, cache eviction policies can be synergistically combined with chunked prefilling to accelerate the pre-filling phase. Following the experimental protocol of DuoAttention (Xiao et al., 2024b), we implemented chunked prefilling with varying chunk sizes on 128K context sequences to achieve prefilling acceleration based on the AdaKV baseline with 50% cache size. By evicting non-essential KV pairs during the sequential processing of chunks, the computational and memory overhead for subsequent chunks is reduced. Table 8 presents the pre-filling latency and corresponding speedup factors across varying chunk sizes. Our method incurs minimal computational overhead compared to the AdaKV baseline. For example, at a chunk size of 30K, our method achieves a latency of 22.31s (1.37x speedup), which is comparable to AdaKV's 22.48s (1.36x speedup). Furthermore, the latency increase remains negligible across all chunk sizes, consistently staying within approximately 0.2 seconds of the baseline. Given the significant quality improvements our method offers over existing baselines, this marginal increase in overhead is negligible.

## E    LIMITATIONS

Our work demonstrates that $L_1$ distance-based perturbation-constrained selection algorithms can effectively enhance the retrieval scores of the original SnapKV and AdaKV. We also evaluated the $L_2$ distance metric and found its performance to be similar to the $L_1$ distance. Future work may explore more sophisticated distance metrics within this framework. In addition, our current approach

Table 8: Comparison of 128K context TTFT in seconds.

| Chunk Size | Full Cache | AdaKV | AdaKV (w/ ours) | Chunk Size | Full Cache | AdaKV | AdaKV (w/ ours) |
|---|---|---|---|---|---|---|---|
| 30k | 30.57 (1.00x) | 22.31 (1.37x) | 22.48 (1.36x) | 90k | 30.53 (1.00x) | 25.99 (1.17x) | 26.17 (1.17x) |
| 50k | 30.54 (1.00x) | 23.46 (1.30x) | 23.63 (1.29x) | 110k | 30.43 (1.00x) | 28.04 (1.09x) | 28.21 (1.08x) |
| 70k | 30.50 (1.00x) | 25.09 (1.22x) | 25.24 (1.21x) | 128k | 29.79 (1.00x) | 29.98 (0.99x) | 30.17 (0.99x) |

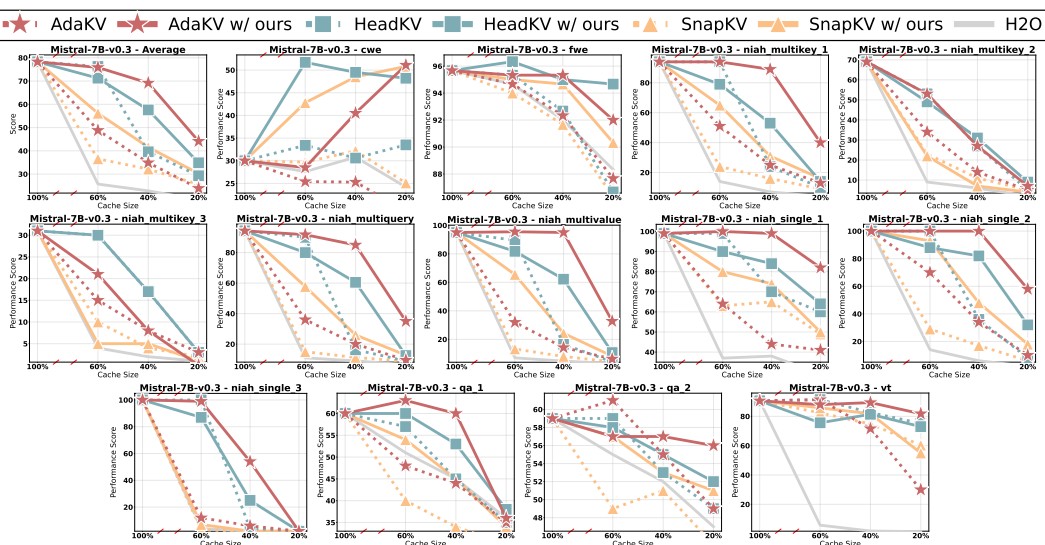

Figure 8: Performance on Ruler Tasks of Mistral-7B-v0.3 with Varying Cache Sizes

assumes that the $\alpha = 50\%$ most important KV cache entries are retained in the first stage to ensure the assumption hold (Appendix A). Nonetheless, exploring more fine-grained strategies can be explored for further improvement.

## F  TASK DOMAIN ANALYSIS OF LONGBENCH RESULTS WITH AN EASY COMPRESSION SETTING

Table 9 reports domain scores on the LongBench benchmark under an easy compression setting, where both the context and question are simultaneously provided for compression . Because this setup allows cache compression targeted to specific questions, it is considered simple and results in minimal quality degradation, with scores nearly matching the full cache case even in 20% cache size. Nevertheless, our enhanced cache eviction method also improves quality across most domains. However, this scenario is not widely applicable in practice, as it fails in multi-turn question answering or real-world contexts where future questions cannot be anticipated. Therefore, we recommend evaluating methods under more challenging compression settings as adopted in our main experiments that better reflect practical use cases.

## G  DETAIL RESULTS OF MISTRAL-7B-v0.3 ON RULER BENCHMARK

Figure 8 presents the detailed results of the Mistral model on the Ruler benchmark with varying cache sizes. Overall, our algorithm significantly improves the performance of all three baseline methods.

## H  DETAILS OF 16 DATASETS IN LONGBENCH

As a widely used long-context benchmark (Feng et al., 2024; Li et al., 2024; Zhang et al., 2024a), LongBench consists of 16 datasets across six task domains: single-document question answering (QA) (Kočiskỳ et al., 2018; Dasigi et al., 2021), multi-document QA (Yang et al., 2018; Ho et al., 2020; Trivedi et al., 2022), summarization (Huang et al., 2021; Zhong et al., 2021; Fabbri et al., 2019), few-shot learning (Joshi et al., 2017; Gliwa et al., 2019; Li & Roth, 2002), synthetic tasks (Bai

Table 9: Domain Scores on LongBench under Easy Compression Setting.

| | Domain | Full Cache | AdaKV $b = 5\%$ | | AdaKV $b = 10\%$ | | AdaKV $b = 20\%$ | | AdaKV $b = 40\%$ | |
|---|---|---|---|---|---|---|---|---|---|---|
| | | | base | w/ ours | base | w/ ours | base | w/ ours | base | w/ ours |
| Llama-3.1-8B Easy Setting | SingleDoc. QA | 43.10 | 38.57 | **38.79** | **41.36** | 41.07 | 42.73 | **43.05** | 43.31 | **43.59** |
| | MultiDoc. QA | 46.49 | 44.61 | **45.28** | 46.03 | **46.08** | **46.64** | 46.42 | **47.02** | 46.97 |
| | Summarization | 28.97 | 22.85 | **22.97** | 24.17 | **24.63** | 25.49 | **26.05** | 27.24 | **27.79** |
| | Fewshot | 69.45 | 67.06 | **67.49** | 68.65 | **68.72** | **69.19** | 69.03 | 69.36 | **69.40** |
| | Synthetic | 53.73 | **53.49** | 53.36 | 53.25 | **53.56** | 53.57 | **54.45** | 53.96 | **54.59** |
| | Code | 57.86 | 56.72 | **57.26** | 57.63 | **58.24** | 58.43 | **58.57** | 58.27 | **58.46** |
| | Ave. Score | 49.20 | 46.23 | **46.55** | 47.65 | **47.82** | 48.51 | **48.73** | 49.08 | **49.33** |
| | Avg. Loss ↓ | 0.0 % | 6.0 % | **5.4** % | 3.2 % | **2.8** % | 1.4 % | **1.0** % | 0.2 % | **-0.3** % |

et al., 2023), and code generation (Guo et al., 2023; Liu et al., 2023). The average token length across all 16 datasets is 6,711. Table 10 provides detailed information on the 16 datasets in LongBench.

Table 10: Details of 16 datasets in LongBench.

| Task | Task Type | Eval metric | Avg len | Language | Sample Num |
|---|---|---|---|---|---|
| NarrativeQA | Single-Doc. QA | F1 | 18,409 | EN | 200 |
| Qasper | Single-Doc. QA | F1 | 3,619 | EN | 200 |
| MultiFieldQA-en | Single-Doc. QA | F1 | 4,559 | EN | 150 |
| HotpotQA | Multi-Doc. QA | F1 | 9,151 | EN | 200 |
| 2WikiMultihopQA | Multi-Doc. QA | F1 | 4,887 | EN | 200 |
| MuSiQue | Multi-Doc. QA | F1 | 11,214 | EN | 200 |
| GovReport | Summarization | Rouge-L | 8,734 | EN | 200 |
| QMSum | Summarization | Rouge-L | 10,614 | EN | 200 |
| MultiNews | Summarization | Rouge-L | 2,113 | EN | 200 |
| TREC | Few-shot Learning | Accuracy | 5,177 | EN | 200 |
| TriviaQA | Few-shot Learning | F1 | 8,209 | EN | 200 |
| SAMSum | Few-shot Learning | Rouge-L | 6,258 | EN | 200 |
| PassageCount | Synthetic | Accuracy | 11,141 | EN | 200 |
| PassageRetrieval-en | Synthetic | Accuracy | 9,289 | EN | 200 |
| LCC | Code | Edit Sim | 1,235 | Python/C#/Java | 500 |
| RepoBench-P | Code | Edit Sim | 4,206 | Python/Java | 500 |

# I  ANALYSIS OF PREVIOUS SOLELY ATTENTION WEIGHTS-BASED SELECTION FROM A PERTURBATION PERSPECTIVE

Our algorithm differs from the previous solely attention weights-based selection method primarily in Stage 2. Specifically, by modifying stage 2 of our algorithm to perform the same attention weights-based selection operation as in stage 1, our approach will degrade into the previous method. This modification allows us to conveniently apply perturbation-constrained theory to analyze the earlier attention weights-based selection strategy.

**Theorem I.1.** *Previous solely attention weights-based selection is equivalent to minimizing another upper bound $\hat{\theta}^{relax}$, a relaxed form of $\hat{\theta}$, with remaining budget $b''$ based on stage 1 selection.*

$$\hat{\theta}^{relax} = C' - M \left( 2 - \frac{1}{\sigma} \right) \sum_{i=1}^{n} \mathcal{N}_i'' A_i \quad where \quad M = MIN(\|\boldsymbol{\mathcal{V}}_{i,:}\|_1) \tag{7}$$

*Proof.* We relax the upper bound $\hat{\theta}$ by utilizing $M = MIN(\|\boldsymbol{\mathcal{V}}_{i,:}\|_1)$:

$$\hat{\theta} = C' - \left( 2 - \frac{1}{\sigma} \right) \sum_{i=1}^{n} \mathcal{N}_i'' A_i \|\boldsymbol{\mathcal{V}}_{i,:}\|_1 \leq C' - M \left( 2 - \frac{1}{\sigma} \right) \sum_{i=1}^{n} \mathcal{N}_i'' A_i = \hat{\theta}^{relax} \tag{8}$$

In the solely attention weights-based selection strategy, the $\mathcal{N}''$ selection is performed using $Top - K(A_i, b'')$ to maximize $\sum_{i=1}^{n} \mathcal{N}_i'' A_i$. This is therefore equivalent to minimizing the relaxed upper bound, $\hat{\theta}^{relax}$.

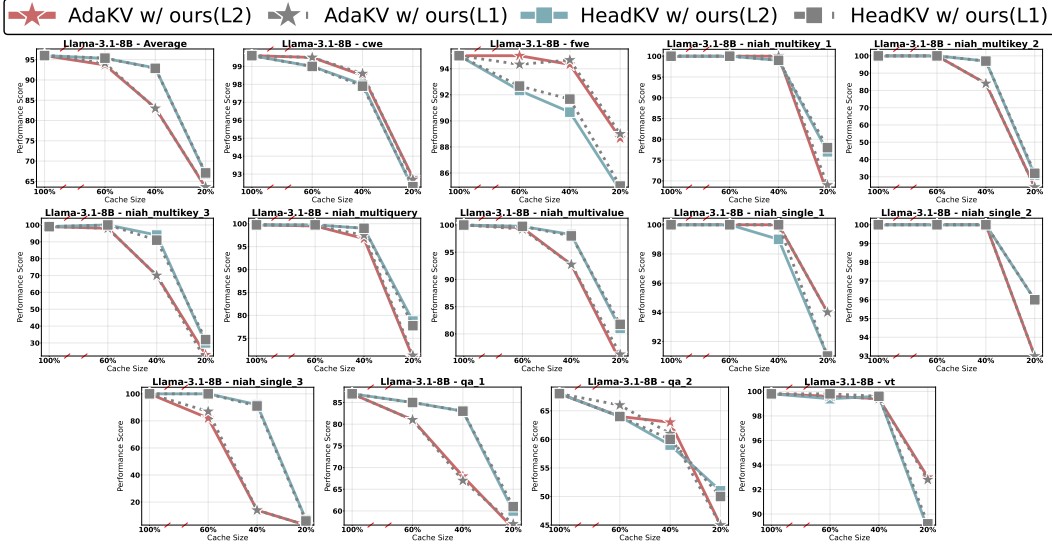

Figure 9: Choice of Distance Metric: $L_1$ distance and $L_2$ distance.

$\square$

Theorem I.1 demonstrates that the solely attention weights-based selection strategy is equivalent to minimizing the relaxed upper bound $\hat{\theta}^{relax}$. In contrast, our algorithm optimizes a tighter upper bound, $\hat{\theta}$. While this does not guarantee that our approach will yield a strictly better solution, intuitively, an algorithm designed to optimize a tighter bound often achieves better results. Theorem I.1 also provides some insight into why a critical KV Cache subset can replace the entire KV Cache in cache eviction methods. Due to the power-law distribution of attention weights (Zhang et al., 2024b), removing most cache entries with near-zero attention weights has a negligible impact on this upper bound. Consequently, the perturbation to the actual output is also bounded by this upper bound.

## J  CHOICE OF DISTANCE METRIC

To evaluate the impact of different distance metrics on our algorithm, we compared the commonly used $L_1$ and $L_2$ distances on the 4K Ruler Benchmark. As shown in Figure 9, we observed no significant improvement in quality when using the more complex $L_2$ distance compared to the simpler $L_1$ distance. For its simplicity, we adopt the $L_1$ distance metric in our analysis. Exploring more advanced distance metrics within our framework remains a promising direction for future work.

## K  PROOFS FOR THEOREMS

### K.1  PROOF FOR THEOREM 3.2

*Theorem.* By introducing a mask $\mathcal{N} \in \mathbb{R}^n$ applied through element-wise multiplication denoted by $\odot$, we can establish the relation between $A'$ and $A$ as follows:

$$A' = \frac{\mathcal{N} \odot A}{\sum_{i=1}^n \mathcal{N}_i A_i} \quad \text{where } \mathcal{N}_i \qquad = \begin{cases} 0 & \text{if } K_i, V_i \text{ is non-critical} \\ 1 & \text{otherwise.} \end{cases} \text{ and } \sum_{i=1}^n \mathcal{N}_i = b$$

*Proof.* Let $a = qK^T/\sqrt{d}$, we can express the attention weights $A'$ under critical cache entries as:

$$A' = \frac{exp(\mathcal{M} + a)}{\sum_{i=1}^n exp(\mathcal{M} + a)_i} \tag{9}$$

$$= \frac{\mathcal{N} \odot exp(a)}{\sum_{i=1}^n \mathcal{N}_i exp(a)_i}$$

$$= \mathcal{N} \odot \frac{exp(a)}{\sum_{i=1}^n exp(a)_i} \frac{\sum_{i=1}^n exp(a)_i}{\sum_{i=1}^n \mathcal{N}_i exp(a)_i}$$

Considering $A = \frac{exp(a)}{\sum_{i=1}^n exp(a)_i}$, thus $\sum_{i=1}^n \mathcal{N}_i A_i = \frac{\sum_{i=1}^n \mathcal{N}_i exp(a)_i}{\sum_{i=1}^n exp(a)_i}$. Therefore, $A' = \frac{\mathcal{N} \odot A}{\sum_{i=1}^n \mathcal{N}_i A_i}$. $\square$

### K.2 PROOF FOR THEOREM 3.3

*Theorem.* The output perturbation $\mathcal{L}$ can be bounded by $\theta$:

$$\mathcal{L} \le \theta = C - \left(2 - \frac{1}{\sum_{i=1}^n \mathcal{N}_i A_i}\right) \sum_{i=1}^n \mathcal{N}_i A_i \|\boldsymbol{\mathcal{V}}_{i,:}\|_1, \tag{10}$$

where $C$ denotes the $\sum_{i=1}^n A_i \|\boldsymbol{\mathcal{V}}_{i,:}\|_1$ and $\boldsymbol{\mathcal{V}} \in \mathbb{R}^{n \times d} = VW^O$ denotes all projected values states through parameter matrix $W^O$.

*Proof.* Let $\boldsymbol{\mathcal{V}} \in \mathbb{R}^{n \times d} = VW^O$ denote all projected value states, thus:

$$\mathcal{L} = \|\left(A - \frac{\mathcal{N} \odot A}{\sum_{i=1}^n \mathcal{N}_i A_i}\right) \boldsymbol{\mathcal{V}}\|_1 \tag{11}$$

$$= \|\sum_{i=1}^n \left(A_i - \frac{\mathcal{N}_i A_i}{\sum_{i=1}^n \mathcal{N}_i A_i}\right) \boldsymbol{\mathcal{V}}_{i,:}\|_1$$

$$\le \theta = \sum_{i=1}^n \|\left(A_i - \frac{\mathcal{N}_i A_i}{\sum_{i=1}^n \mathcal{N}_i A_i}\right) \boldsymbol{\mathcal{V}}_{i,:}\|_1 \tag{12}$$

$$= \sum_{i=1}^n |A_i - \frac{\mathcal{N}_i A_i}{\sum_{i=1}^n \mathcal{N}_i A_i}| \times \|\boldsymbol{\mathcal{V}}_{i,:}\|_1$$

Given that the multiplicative mask $\mathcal{N}$ is either 0 or 1, the index set $i \in [1, n]$ can be split into $I_0$ and $I_1$, according to its value. Thus:

$$\theta = \sum_{i \in I_0} A_i \|\boldsymbol{\mathcal{V}}_{i,:}\|_1 + \sum_{i \in I_1} \left(\frac{A_i}{\sum_{i=1}^n \mathcal{N}_i A_i} - A_i\right) \|\boldsymbol{\mathcal{V}}_{i,:}\|_1 \tag{13}$$

Let $C$ represent $\sum_{i=1}^n A_i \|\boldsymbol{\mathcal{V}}_{i,:}\|_1$, a constant independent of the selection of critical entries. We can express $\sum_{i \in I_0} A_i \|\boldsymbol{\mathcal{V}}_{i,:}\|_1$ as $C - \sum_{i \in I_1} A_i \|\boldsymbol{\mathcal{V}}_{i,:}\|_1$. Thus:

$$\mathcal{L} \le \theta = C + \sum_{i \in I_1} \left(\frac{A_i}{\sum_{i=1}^n \mathcal{N}_i A_i} - 2A_i\right) \|\boldsymbol{\mathcal{V}}_{i,:}\|_1 \tag{14}$$

$$= C - \left(2 - \frac{1}{\sum_{i=1}^n \mathcal{N}_i A_i}\right) \sum_{i=1}^n \mathcal{N}_i A_i \|\boldsymbol{\mathcal{V}}_{i,:}\|_1$$

$\square$

### K.3 PROOF FOR THEOREM 3.5

*Theorem.* Given the stage 1 selection $\mathcal{N}_i'$, the objective $\mathcal{N}_i''$ of stage 2 is to minimize an upper bound $\hat{\theta}$ of the output perturbation $\mathcal{L}$, using the remaining budget $b'' = b - b'$.

$$\arg\min_{\mathcal{N}_i''} \hat{\theta} \text{ where } \hat{\theta} = C' - \left(2 - \frac{1}{\sigma}\right) \sum_{i=1}^n \mathcal{N}_i'' A_i \|\boldsymbol{\mathcal{V}}_{i,:}\|_1$$

$$\text{subject to } \sum_{i=1}^n \mathcal{N}_i'' = b'', C' = C - \left(2 - \frac{1}{\sigma}\right) \sum_{i=1}^n \mathcal{N}_i' A_i \|\boldsymbol{\mathcal{V}}_{i,:}\|_1. \tag{15}$$

*Proof.* From Assumption 3.4, the first stage selection ensures: $\sum_{i=1}^{n} \mathcal{N}_i A_i > \sum_{i=1}^{n} \mathcal{N}'_i A_i = \sigma > 0.5$, leading to the inequality: $2 - \frac{1}{\sum_{i=1}^{n} \mathcal{N}_i A_i} > 2 - \frac{1}{\sigma} > 0$.

$$
\begin{aligned}
\theta =& C - \left(2 - \frac{1}{\sum_{i=1}^{n} \mathcal{N}_i A_i}\right) \sum_{i=1}^{n} (\mathcal{N}'_i + \mathcal{N}''_i) A_i \|\boldsymbol{\mathcal{V}}_{i,:}\|_1 \\
<& C - \left(2 - \frac{1}{\sigma}\right) \sum_{i=1}^{n} \mathcal{N}'_i A_i \|\boldsymbol{\mathcal{V}}_{i,:}\|_1 \\
& - \left(2 - \frac{1}{\sigma}\right) \sum_{i=1}^{n} \mathcal{N}''_i A_i \|\boldsymbol{\mathcal{V}}_{i,:}\|_1
\end{aligned}
\tag{16}
$$

Let $C' = C - \left(2 - \frac{1}{\sigma}\right) \sum_{i=1}^{n} \mathcal{N}'_i A_i \|\boldsymbol{\mathcal{V}}_{i,:}\|_1$, then we can derive a new upper bound $\hat{\theta}$ for $\mathcal{L}$ factoring by second stage selection $\mathcal{N}''_i$: $\theta < C' - \left(2 - \frac{1}{\sigma}\right) \sum_{i=1}^{n} \mathcal{N}''_i A_i \|\boldsymbol{\mathcal{V}}_{i,:}\|_1 = \hat{\theta}$ Thus, minimizing $\hat{\theta}$ corresponds to selecting the $b''$ entries with the highest values of $\boldsymbol{\mathcal{A}}_i = A_i \|\boldsymbol{\mathcal{V}}i,:\|_1$, as implemented in the stage 2 selection (Algorithm 1). $\square$

## L  ADDITIONAL RELATED WORKS

Some adaptive methods in KV cache eviction or sparse attention, such as (Ge et al., 2024b; Jiang et al., 2024), employ varying critical cache selection strategies tailored to the characteristics of different attention heads. For example, some heads use attention weights based selection, while others utilize fixed patterns, such as recent window-based or special token-based approaches. Our method can also be applied to enhance performance in the head which according to attention weights-based selection strategies, providing a boost to adaptive methods.

A range of techniques beyond cache eviction have also been explored to reduce the KV cache footprint. Think (Xu et al., 2024) compresses the cache by decreasing the number of channels in key states. Methods like MiniCache exploit similarities between layers to achieve compact representations (Liu et al., 2024a; Yang et al., 2024b). KV cache quantization (Liu et al., 2024c; Hooper et al., 2024) also contributes by lowering the precision of individual entries. All of these methods are orthogonal to cache eviction and offer potential for further enhancement.

Dynamic Cache Selection methods (Jiang et al., 2024; Tang et al., 2024b; Lv et al., 2024; Zhang et al., 2025a), such as quest, are conceptually related to the KV cache eviction methods discussed in this paper. While KV cache eviction retains only a small subset of essential KV cache entries, sparse attention methods maintain all entries during inference. However, during computation, only the most critical entries are selectively utilized in the sparse attention mechanism. Consequently, sparse attention methods do not reduce the memory footprint of the KV cache but enhance inference speed and often offer better output quality than cache eviction methods (Tang et al., 2024b). Existing sparse attention methods typically rely on approximate estimations of attention weights to identify critical entries (Tang et al., 2024b; Lv et al., 2024). Future works could explore integrating our proposed perturbation-constrained selection algorithm to refine these methods by achieving more accurate critical cache entry identification.

## M  PROMPT TEMPLATES FOR RULER AND LONGBENCH IN REGULAR AND CONTEXT-ONLY COMPRESSION SCENARIOS

Below are prompt templates for various tasks. We assess performance under two scenarios: regular compression and context-only compression. We adhere to the input prompt format from KVPress (NVIDIA, 2024), dividing the input into context and question segments. The question segment is highlighted in green, while other colors represent the context segment. In regular compression, both the context and question segments are input into the model and compressed. For context-only compression, where future questions are unpredictable, only the context segment is input for compression. After compression, the question segment is input for answer generation.

### M.1  NIAH TEMPLATE

In the Needle-in-A-Haystack task, a keyword, referred to as the "needle", is embedded within a lengthy context known as the "haystack". The objective of this task is to extract the "needle" from the "haystack", which is composed of essays by Paul Graham (Kamradt, 2023).

For the Single Needle-in-A-Haystack(S-NIAH) task, the goal is to retrieve a single "needle". Similarly, the Multi-Value Needle-in-A-Haystack(MV-NIAH) task requires the extraction of multiple inserted "needles". To prevent models from refusing to answer our questions, we append the answer prefix to the input, prompting the models to generate answers.

Table 11: Single retrieval and multi retrieval templates in Needle-in-A-Haystack tests.

| | |
|---|---|
| Single retrieval | **Task Template:**
Some special magic numbers are hidden within the following text. Make sure to memorize it. I will quiz you about the numbers afterwards.
Paul Graham Essays.
...... One of the special magic numbers for {word} is: {number}. ......
What is the special magic number for {word} mentioned in the provided text?

The special magic number for {word} mentioned in the provided text is |
| Multi retrieval | **Task Template:**
Some special magic numbers are hidden within the following text. Make sure to memorize it. I will quiz you about the numbers afterwards.
Paul Graham Essays.
...... One of the special magic numbers for {word} is: {number-1}. ......
...... One of the special magic numbers for {word} is: {number-2}. ......
...... One of the special magic numbers for {word} is: {number-3}. ......
...... One of the special magic numbers for {word} is: {number-4}. ......
What are all the special magic numbers for {word} mentioned in the provided text?

The special magic numbers for {word} mentioned in the provided text are |

## M.2 LONGBENCH TEMPLATE

The construction of the LongBench template follows the official formats (Bai et al., 2024) to evaluate performance under regular compression and context-only compression.

Table 12: LongBench templates. Single-Doc. QA Tasks.

| | |
|---|---|
| NarrativeQA | **Task Template:**
You are given a story, which can be either a novel or a movie script, and a question. Answer the question asconcisely as you can, using a single phrase if possible. Do not provide any explanation.

Story: {context}

Now, answer the question based on the story asconcisely as you can, using a single phrase if possible. Do not provide any explanation.

Question: {question} |
| Qasper | **Task Template:**
You are given a scientific article and a question. Answer the question as concisely as you can, using a single phrase or sentence if possible. If the question cannot be answered based on the information in the article, write "unanswerable". If the question is a yes/no question, answer "yes", "no", or "unanswerable". Do not provide any explanation.

Article: {context}

Answer the question based on the above article as concisely as you can, using a single phrase or sentence if possible. If the question cannot be answered based on the information in the article, write "unanswerable". If the question is a yes/no question, answer "yes", "no", or "unanswerable". Do not provide any explanation.

Question: {question} |
| MultifieldQA EN | **Task Template:**
Read the following text and answer briefly.

{context}

Now, answer the following question based on the above text, only give me the answer and do not output any other words.

Question: {question} |

Table 13: LongBench templates. Multi-Doc. QA Tasks.

| | |
|---|---|
| HotpotQA | **Task Template:**
Answer the question based on the given passages. Only give me the answer and do not output any other words.

The following are given passages.
{context}

Answer the question based on the given passages. Only give me the answer and do not output any other words.

Question: {question} |
| 2WikimQA | **Task Template:**
Answer the question based on the given passages. Only give me the answer and do not output any other words.

The following are given passages.
{context}

Answer the question based on the given passages. Only give me the answer and do not output any other words.

Question: {question} |
| Musique | **Task Template:**
Answer the question based on the given passages. Only give me the answer and do not output any other words.

The following are given passages.
{context}

Answer the question based on the given passages. Only give me the answer and do not output any other words.

Question: {question} |

Table 14: LongBench templates. Summarization Tasks.

| | |
|---|---|
| Gov Report | **Task Template:**
You are given a report by a government agency. Write a one-page summary of the report.

Report:
{context}

Now, write a one-page summary of the report. |
| QMSum | **Task Template:**
You are given a meeting transcript and a query containing a question or instruction. Answer the query in one or more sentences.

Transcript:
{context}

Now, answer the query based on the above meeting transcript in one or more sentences.

Query: {question} |
| Multi News | **Task Template:**
You are given several news passages. Write a one-page summary of all news.

News:
{context}

Now, write a one-page summary of all the news. |

Table 15: LongBench templates. Few-shot Learning Tasks.

| | |
|---|---|
| TREC | **Task Template:**
Please determine the type of the question below. Here are some examples of questions.

{context}
{question} |
| TriviaQA | **Task Template:**
Answer the question based on the given passage. Only give me the answer and do not output any other words. The following are some examples.

{context}

{question} |
| SAMSum | **Task Template:**
Summarize the dialogue into a few short sentences. The following are some examples.

{context}

{question} |

Table 16: LongBench templates. Synthetic Tasks.

| | |
|---|---|
| Passage Count | **Task Template:**
There are some paragraphs below sourced from Wikipedia. Some of them may be duplicates. Please carefully read these paragraphs and determine how many unique paragraphs there are after removing duplicates. In other words, how many non-repeating paragraphs are there in total?

{context}

Please enter the final count of unique paragraphs after removing duplicates. The output format should only contain the number, such as 1, 2, 3, and so on. |
| Passage Retrieval EN | **Task Template:**
Here are 30 paragraphs from Wikipedia, along with an abstract. Please determine which paragraph the abstract is from.

{context}

The following is an abstract.

{question}

Please enter the number of the paragraph that the abstract is from. The answer format must be like "Paragraph 1", "Paragraph 2", etc. |

Table 17: LongBench templates. Code Tasks.

| | |
|---|---|
| Lcc | **Task Template:**
Please complete the code given below.
{context}
Next line of code: |
| Repobench-P | **Task Template:**
Please complete the code given below.
{context}
{question}
Next line of code: |

