# OpenReview forum: "Identify Critical KV Cache in LLM Inference from an Output Perturbation Perspective"
_ICLR.cc/2026/Conference — Submitted to ICLR 2026_

### Official Review · Reviewer_FGK5 · 2025-10-18

**Soundness:** 1
**Presentation:** 2
**Contribution:** 2
**Rating:** 2
**Confidence:** 5

**Summary:**

This paper proposes a plug-and-play method to enhance existing KV cache eviction strategies. Concretely, it heuristically sets a fraction factor α = 0.5: 50% of the KV budget is evicted using an existing method, and the remaining budget is selected based on A*|V|_1 to approximate output perturbation importance. Experiments on LongBench and Ruler show some improvements over baselines.

**Strengths:**

1. This paper is generally well-written.
2. The performance improvements over baseline methods on Longbench and Ruler are notable.

**Weaknesses:**

1) The paper claims that recent efforts are empirical and lack formal grounding, and that this work presents a formal study from the perspective of output perturbation. However, reasoning from output error is the most straightforward approach; the reason the field often avoids it is the high computational cost of evaluating A*V. The proposed “formal” method in practice is a heuristic: keep half of the budget with an existing method and allocate the rest via a relaxed A*|V|_1 criterion. Moreover, prior methods such as Quest and MagicPig also consider output-level effects.

2) Let's go further from last point and discuss the soundness of the proposed method. Quest also uses an approximate A*V to select important tokens, but it is more justified because it makes on-the-fly decisions during inference, step-by-step, conditioned on each incremental output (I'm not asking the authors to compare Quest since its a selection method). By contrast, this paper’s formulation is built around the output of a single token at the end of prefilling. That is naturally aligned with single-turn QA settings like LongBench and ruler where most information resides in the prompt, but it is too narrow for real-world scenarios. In my view, such output-perturbation–based scoring is better suited for cache selection rather than eviction. The paper should include results on datasets like PG-19 and SCBench that feature multi-turn dialogue or long-horizon modeling.

3) Efficiency evaluation is unconvincing. The paper uses time-to-first-token (TTFT) in the prefill phase (including eviction) as the efficiency metric. Under the current evaluation setting, where full prefilling is followed by a one-shot eviction based on some metric, then decoding with the remaining KV, I think the TTFT does not make sense at all.  With one-shot removal, the complexity of the scoring method hardly matters; one could even use a full A*V and see little difference in TTFT compared to what is reported. What would be meaningful is: (a) chunk-wise acceleration during prefilling, or (b) on-the-fly acceleration during long-context decoding. At minimum, please quantify the latency/memory overhead per decoding step for the proposed approximate selection, since Quest found such approximations to be costly and thus opted for block-wise computation.

4) Clarity and presentation. The paper is not that easy to follow. The method is simple enough to be described clearly in a few paragraphs as the core idea is to introduce an approximate A*V to guide the first-step output perturbation for cache eviction. The mathematical proofs seem unnecessary.

5) Naming. The method needs a proper name. Referring to it as “ours” throughout the paper is unhelpful for readers and subsequent citations.

**Questions:**

Please see the above part.

---

> ### Author Response · Authors · 2025-11-22
> **Reply to Reviewer (Part 1)**
>
> Thank you for your effort.
>
> **We would first like to clarify the scope of our work.**  As detailed in the table below, Static Cache Eviction and Dynamic Cache Selection are orthogonal domains with distinct objectives and challenges. Moreover, they are complementary in practice: one can first apply Cache Eviction to reduce the memory footprint, and subsequently use Dynamic Cache Selection for further acceleration.
>
> As stated in the Abstract and Introduction, **our contribution** is positioned as using formal, perturbation-based analysis to improve existing **Static Cache Eviction** methods. While the framework could in principle be extended, this is beyond our focus and would require addressing setting-specific challenges—for example, the per-step overhead issues you mentioned for dynamic cache selection.
>
> **We are open to adding “Cache Eviction” as a keyword in the title if this would help make the scope clearer.**
>
> | Category | Static Cache Eviction|Dynamic Cache Selection|
> |-|-|-|
> |Representative Methods|**Ours**/SnapKV/AdaKV/HeadKV|Quest/MagicPig/MInference|
> |Operation Frequency| **Once** post-prefilling per layer to reduces peak memory.| **Per-step** dynamic selection for speedup. |
> | Per-Step Perturbation Modeling|**No.**  Infeasible for single-shot static compression. |**Yes.** Requires on-the-fly modeling at each step.|
> | Memory Saving | **Yes.** Permanently discards tokens, effectively reducing the VRAM footprint. | **No.** Typically retains the full cache in VRAM to support on-the-fly selection. |
> | Decoding Speedup | **Yes.** Accelerates attention computation via a reduced cache size.| **Yes.** Reduces latency by skipping computation and I/O for non-essential cache. |
> | Prefilling Speedup | **No.** Generally requires full computation (unless combined with chunked prefill). | **Yes.** Can sparsify computation directly during the prefilling phase. |
> |Primary Scope| 	**Memory-Constrained** Scenarios. Ideal for long-context serving on limited VRAM; enables larger batch sizes to greater throughput.| **Latency-Sensitive** Scenarios. Focuses on reducing inference latency by optimizing I/O and FLOPs, without reducing peak memory usage.|
> ════════════════════
> ## Reply to Weakness 1：
>
> > However, reasoning from output error is  straightforward. The reason the field often avoids it is the high computational cost of evaluating AV.
>
> **We respectfully disagree with the premise regarding computational cost in our context and offer empirical evidence showing that directly using $A|V|$ is ineffective.**
>
> As mentioned above,  in the **static cache eviction** addressed in this paper, evaluating $A|V|$ does not incur a high computational cost. This is substantiated by our efficiency experiments on augmenting existing cache eviction methods. While this costs might hinder the application in **dynamic cache selection**, those scenarios are **outside the scope of this work.**
>
> Crucially, **directly using \(A|V|\) is ineffective.** As evidence, we implement a variant of the AdaKV-based KV cache eviction algorithm that **directly uses \(A|V|\) as the selection criterion, which actually performs even worse than AdaKV**:
>
> |Longbench Avg. (16 datasets) | Llama 20% Cache | Mistral 20% Cache |
> |-|-|-|
> |Full Cache| 49.20 | 47.38 |
> |AdaKV|42.87(12.9%↓) | 41.78(11.8%↓) |
> |AdaKV w/ AV|42.51(13.6%↓) | 41.8(11.8%↓) |
> |AdaKV w/ ours|**43.77(11.0%↓)** | **42.85(9.6%↓)** |
>
> The superiority of our approach stems from two key factors. **First,  our algorithm incorporates the pre-trained parameter matrix $W^O$, capturing information beyond simple value states.** Second, **we adopt a two-stage selection strategy**: the first stage allocates a "safeguard" budget based on Assumption 3.4, which ensures that the second stage can effectively integrate $A|VW^O|$ to constrain the theoretical upper bound of the perturbation.
>
> > The proposed “formal” method in practice is a heuristic.
>
> We may be misunderstanding the concern, but our intent is to present a theory-gudied two-stage algorithm:
>
> - The first stage acts as a safeguard for Assumption 3.4.
>
> - The second stage combines the A|VW^O| criterion to theoretically bound perturbation.
>
> Given that optimal selection is computationally intractable, **a theoretically motivated greedy scheme is a standard and formal design choice**, and our ablation studies further validate both its theoretical soundness and its empirical performance. We therefore believe the algorithm is substantially more than a heuristic.
>
> > Moreover, prior methods such as Quest and MagicPig also consider output-level effects.
>
> - **Quest** does not provide a theoretical analysis at the output level.
>
> - **MagicPig** focuses on oracle-based sampling and proves the unbiasedness of its sampling estimator, but its theoretical analysis **does not address output perturbation bounds**.
>
> **If we have overlooked some section of these works that explicit guarantees on output perturbation, we would be grateful if reviewer can direct us to it.**

---

> ### Author Response · Authors · 2025-11-22
> **Reply to Reviewer (Part 2)**
>
> ## Reply to Weakness 2:
>
> > Quest also uses an approximate A*V to select important tokens.....
>
> **We respectfully disagree: neither Quest nor MagicPig uses A*V; both methods operate entirely on approximated attention weights.**
>
> > By contrast, this paper’s formulation is built around the output of a single token at the end of prefilling. That is naturally aligned with single-turn QA settings like LongBench and ruler where most information resides in the prompt, but it is too narrow for real-world scenarios. In my view, such output-perturbation–based scoring is better suited for cache selection rather than eviction.
>
>
> We clarify that in the **static cache eviction** setting, explicitly modeling future output perturbation is generally infeasible, because the future questions are unknown at the time of compression， and questions are answered using the compressed cache. Under this setting, **our algorithm naturally extends to multi-turn QA scenarios**, where the compressed context must support multiple unknown questions， This demonstrates that our approach is not narrowly confined to single-turn settings.
>
> Furthermore, in the static eviction setting where operation is infrequent (typically performed once), the computational cost of an output-perturbation score is not a concern. In contrast, extending this idea to dynamic cache selection, per-token scoring for cache selection would incur additional overhead, which is beyond the scope of this work.
>
>
>
>
>
> > The paper should include results on datasets like PG-19 and SCBench that feature multi-turn dialogue or long-horizon modeling.
>
> **As mentioned above, our evaluation already simulates future-question-unknown settings as in multi-turn QA settings.**
>
> In response to your suggestion, we evaluated our method on the multi-turn dialogue benchmark SCBench. **SCBench contains 4,853 QA turns with an average length of 227K tokens, which not only far exceeds what academic research can typically afford, but also surpasses the context capacity of open-source LLMs.**
>
> To meet the constraints of 128K window of Llama-3.1-8B for multi-turn evaluation, we followed established practice and selected three representative tasks, Retr.KV, EN.QA, and Math.Find, from the three categories of String Retrieval, Semantic Retrieval, and Global Information, respectively, truncating the context to 100K tokens.
>
>
> | Domain              | Full Cache | 80% Cache AdaKV | 80% Cache AdaKV (w/ ours) | 60% Cache AdaKV | 60% Cache AdaKV (w/ ours) | 40% Cache AdaKV | 40% Cache AdaKV (w/ ours) |
> |-|-|--|--|-|-|--|-|
> | Retr.KV (turns=5)   | 52.2       | 52.00           | **52.6**                  | 41.2            | **42**                    | 19.4            | **19.8**                  |
> | Math.Find (turns=6) | 11.67      | 11.33           | **13.67**                 | 11.67           | **14.17**                 | 11.5            | **16.83**                 |
> | EN.QA (turns=7)     | 22.86      | 22.14           | **24.29**                 | 17.86           | **21.43**                 | 15.71           | **22.14**                 |

---

> ### Author Response · Authors · 2025-11-22
> **Reply to Reviewer (Part 3)**
>
> ## Reply to Weakness 3:
>
> > Efficiency evaluation is unconvincing. The paper uses time-to-first-token (TTFT) in the prefill phase (including eviction) as the efficiency metric. Under the current evaluation setting, where full prefilling is followed by a one-shot eviction based on some metric, then decoding with the remaining KV, I think the TTFT does not make sense at all.
>
> We respectfully disagree. Cache eviction methods naturally operate as described, and our work seeks to improve their performance. **This TTFT-based metric is commonly used for cache eviction and can be found in many similar studies, including our baselines AdaKV and HeadKV.**
>
>
> >  With one-shot removal, the complexity of the scoring method hardly matters; one could even use a full A*V and see little difference in TTFT compared to what is reported.
>
> We do not see why the complexity of the evaluation method would be irrelevant.
>
> **In fact, we use the full A*V, confirming its efficiency in cache eviction scenarios, and this approach is appropriate for cache eviction.** Previous work in this area has consistently followed a one-shot removal strategy and has always reported TTFT, **because certain methods indeed significantly increase TTFT.** In some cases, the compression time even far exceeds the prefilling time itself, as in the concurrent NeurIPS 2025 work KVzip.
>
> KVzip: Query-Agnostic KV Cache Compression with Context Reconstruction. Neurips 2025
>
>
> > What would be meaningful is: (a) chunk-wise acceleration during prefilling, or (b) on-the-fly acceleration during long-context decoding.  At minimum, please quantify the latency/memory overhead per decoding step for the proposed approximate selection, since Quest found such approximations to be costly and thus opted for block-wise computation.
>
> To ensure high-quality generation, cache eviction methods typically maintain the integrity of the prefilling phase; chunk-wise acceleration compromises this. In practice, cache eviction is typically applied after the prefilling phase of each layer to reduce memory footprint efficiently. If speed results are of primary interest, we can present them.
>
> 128K TTFT(s) with Chunked Prefill
> | Chunk Size | Full Cache | AdaKV (50% Cache) | AdaKV w/ ours (50% Cache ) |
> |-|--|-|--|
> | 10k | 30.88(1.00x)   | 21.57(1.43x) | 21.72(1.42x)    |
> | 30k | 30.57(1.00x)   | 22.31(1.37x) | 22.48(1.36x)    |
> | 50k | 30.54(1.00x) | 23.46(1.30x) | 23.63(1.29x)    |
> | 70k | 30.50(1.00x)   | 25.09(1.22x) | 25.24(1.21x)    |
> | 90k | 30.53(1.00x)  | 25.99(1.17x) | 26.17(1.17x)    |
> | 110k | 30.43(1.00x)  | 28.04(1.09x) | 28.21(1.08x)    |
> | 128k | 29.79(1.00x) | 29.98(0.99x) | 30.17(0.99x)    |
>
>
>
> Our work, as stated in the introduction and abstract, **focuses on improving static cache eviction, not on-the-fly acceleration like Quest**. Extending to dynamic cache selection is feasible but outside the scope of this paper. If necessary, we are open to including “cache eviction” in the title for clarity.
>
> ## Reply to Weakness 4：
>
> > Clarity and presentation. The paper is not that easy to follow. The method is simple enough to be described clearly in a few paragraphs as the core idea is to introduce an approximate A*V to guide the first-step output perturbation for cache eviction. The mathematical proofs seem unnecessary.
>
> **As discussed above, directly using A*V is insufficient and can even degrade performance, so the method cannot be described as "simply using A*V".**  Our contribution lies in a **theoretically grounded two-stage algorithm** that explains why naive A*V fails and why our algorithm works. The theoretical components are essential for motivating the design.
>
> If preferred, we can move the theoretical details to the appendix and keep the main text focused on the algorithm.
>
>
>
> ## Reply to Weakness 5：
>
> > Naming. The method needs a proper name. Referring to it as “ours” throughout the paper is unhelpful for readers and subsequent citations.
>
> Thank you for the suggestion. We will name the three variants of our method as CriticalKV, AdaCriticalKV, and HeadCriticalKV, and will update these terms accordingly throughout the revised manuscript.
>
>
> ═══════════════════════════════════════════════════════════════════════
>
> **Overall, our work is centered on the cache eviction domain. Accordingly, our choices regarding dataset selection, compression settings, and the reporting of both quality and efficiency are consistent with common practices in prior work. If our clarifications do not fully address your concerns, we welcome suggestions for more specific evaluation methods relevant to cache eviction—for example, results comparing particular datasets, baselines, or specific settings, which we would be happy to provide.**

---

### Official Review · Reviewer_FWKD · 2025-10-23

**Soundness:** 3
**Presentation:** 3
**Contribution:** 3
**Rating:** 8
**Confidence:** 3

**Summary:**

This paper presents a formal and theoretically grounded framework for identifying critical Key-Value (KV) cache entries during large language model (LLM) inference. Unlike prior empirical cache-eviction methods that depend solely on attention weights (e.g., H2O, SnapKV, AdaKV, HeadKV), the authors analyze the output perturbation caused by cache compression.
They derive an upper bound on the worst-case perturbation (Theorem 3.3), showing that both attention weights and projected value states (through $W_O$) are essential.
Based on this insight, they design a two-stage perturbation-constrained selection algorithm that can be plugged into existing eviction frameworks.
Experiments on Ruler (13 synthetic long-context tasks) and LongBench (16 real-world datasets) demonstrate consistent performance gains—reducing compression loss by more than half across three models (Llama-3.1-8B, Mistral-7B, Qwen2.5-32B) and three cache-eviction baselines.
The method adds negligible computational overhead.

**Strengths:**

1. The authors are the first to formalize KV cache criticality through output perturbation analysis. The derivation of an upper bound provides solid theoretical insight missing in prior work.
2. The algorithm integrates seamlessly with multiple state-of-the-art eviction methods (SnapKV, AdaKV, HeadKV) and consistently improves them without modifying their architectures.
3. Extensive experiments on 29 datasets across 3 model families (7B–32B scale) show robust improvements, with detailed ablations over cache sizes, layers, and heads.

**Weaknesses:**

1. Although Appendix B claims robustness to α, the main paper only presents results for α = 0.5. A sensitivity analysis in the main text would improve credibility.
2. The writing is mostly clear but sometimes verbose. Sections 3.4–3.6 mix implementation and theoretical statements, which could be reorganized for readability.

**Questions:**

How sensitive is the method to α across very small cache budgets (<10%)?

---

> ### Author Response · Authors · 2025-11-22
> **Reply to Reviewer**
>
> Thank you for your thoughtful review and the effort you invested in evaluating our work. We appreciate your recognition of our solid theoretical insights into KV cache criticality, the plug-and-play nature of our algorithm with existing SOTA methods, and our comprehensive experimental validation across multiple model families. We provide detailed responses to your concerns below.
>
> ## Reply to Weakness 1& Question 1: A sensitivity analysis in the main text would improve credibility. & How sensitive is the method to α across very small cache budgets (<10%)?
>
>
> Thank you for your suggestion. We have added this discussion to Section 4.5 of main paper. We further perform a sensitivity analysis under a 10% cache setting, and include the results into appendix.
>
>
> The results demonstrate that our method remains robust to the choice of $\alpha$, even when the small cache size which causes significant compression loss. The default setting ($\alpha=0.5$) yields substantial improvements over naive AdaKV, and other values (0.3 and 0.7) also show consistent benefits. Conversely, setting $\alpha=0$ may lead to performance degradation due to the violation of our underlying assumption, which is consistent with our previous robustness analysis.
>
>
>
>
> | Longbench  Ave. Llama 10% Cache | Multi_doc | Single_doc | Summarization | Few_shot | Synthetic | Code | avg |
> |---|---|---|---|---|---|---|---|
> | Full Cache (Upper bound)| 46.49 | 43.10 | 28.97 | 69.45 | 53.73 | 57.86 | 49.20 |
> |---|---|---|---|---|---|---|---|
> | AdaKV| 26.44 | 24.75 | 21.75 | 60.74 | 27.91 | 60.69 | 36.14(26.5%↓) |
> | AdaKV w/ours  alpha=0.3 | 29.88 | 26.71 | 22.67 | 61.39 | 35.95 | 60.46 | 38.43(21.9%↓) |
> | AdaKV w/ours  alpha=0.5 | 30.21 | 27.05 | 22.8 | 61.35 | 35.44 | 60.44 | **38.5(21.7%↓)** |
> | AdaKV w/ours  alpha=0.7 | 29.31 | 26.83 | 22.54 | 61.17 | 31.79 | 60.48 | 37.75(23.3%↓) |
>
> | Longbench Ave.   Mistral 10% Cache | Multi_doc | Single_doc | Summarization | Few_shot | Synthetic | Code | avg |
> |---|---|---|---|---|---|---|---|
> | Full Cache (Upper bound)| 39.40 | 38.37 | 28.76 | 70.33 | 52.50 | 61.25 | 47.38 |
> |---|---|---|---|---|---|---|---|
> | AdaKV | 27.56 | 22.78 | 22.14 | 62.24 | 34.0 | 61.77 | 37.23(21.4%↓) |
> | AdaKV w/ours  alpha=0.3 | 25.08 | 24.49 | 22.85 | 63.5 | 33.15 | 60.86 | 37.24(21.4%↓) |
> | AdaKV w/ours  alpha=0.5 | 28.79 | 25.33 | 22.72 | 63.35 | 32.64 | 60.57 | **37.94(19.9%↓)** |
> | AdaKV w/ours  alpha=0.7 | 28.7 | 24.65 | 22.6 | 62.38 | 33 | 60.51 | 37.62(20.6%↓) |
>
> ## Reply to Weakness2：
>
> > The writing is mostly clear but sometimes verbose. Sections 3.4–3.6 mix implementation and theoretical statements, which could be reorganized for readability.
>
> Thank you for your suggestions. We will restructure these sections in the revised manuscript to enhance clarity and readability.
>
>
> ═══════════════════════════════════════════════════════════════════════
>
> **We appreciate your acceptance recommendation and your recognition of our work's value. We hope the above discussion resolves your concerns.**

---

### Official Review · Reviewer_NKA7 · 2025-10-29

**Soundness:** 3
**Presentation:** 3
**Contribution:** 2
**Rating:** 6
**Confidence:** 3

**Summary:**

This paper first argues that many existing methods reduce the KV cache size by pruning or evicting KV entries that are considered to be less important, relying on attention weights. In contrast, this paper proposes a new perspective: the output perturbation should be the gold metric.  Based on this idea,  the paper aims to find a set of b KV entries, under a given cache budget B, to minimize the output disturbance. Theoretically, this paper shows that it is suboptimal to select key items only depending on attentions, and the influence of V and W^O must be considered at the same time. A two-stage greedy algorithm is proposed to find the critical KV, which is also a plug-and-play module, with negligible computational overhead.

**Strengths:**

1. Experiments confirm that the algorithm can effectively reduce the output disturbance and provide good performance under different cache sizes.

2. Good writing and illustrations.

3. Comprehensive experiments.

**Weaknesses:**

1. Some kV entries may be important to maintain long-distance dependency or semantic coherence, even if they contribute little to the l_1 loss of the current output. Could you provide some discussion about this point?

2. The greedy strategy does not guarantee finding the global optimal solution. Could you provide some output L1 result comparisons between the greedy strategy and exhaustive search? Is it possible?

3. The conclusion of the output difference relies on attention weight, V, and W^O is straightforward. However, this article has spent a lot of space proving this, which makes me puzzled.

4. Why must this paper use 50% of the budget to select the KV pair with maximum attention weights? Why not use the adjusted attention containing output differences directly (In Alg. 1)?

**Questions:**

Please see Weaknesses.

I think this paper is a good paper and the overal logic are self-motivated.

---

> ### Author Response · Authors · 2025-11-22
> **Reply to Reviewer**
>
> Thank you for your thoughtful review and the effort you invested in evaluating our work. We sincerely appreciate your recognition of our contributions, particularly that the overall logic is self-motivated and the effectiveness is grounded in comprehensive experiments. Below we address your concerns.
>
>
> ## Reply to W1: Some kV entries may be important, even if they contribute little to the l_1 loss of the current output...
>
> We agree with your insightful point. A small L1 perturbation could sometimes lead to a disproportionate drop in quality. Due to the limited interpretability in NNs, devising an oracle metric for such effects is highly challenging.
> However, **our empirical results consistently show that smaller perturbations strongly correlate with smaller quality degradation, which validates the usefulness of our proposal.**
>
> Looking ahead, **we believe the key is to develop perturbation metrics beyond basic $L_1$ or $L_2$ norms toward oracle-like fine-grained perturbation metrics.** Such metrics would enable more accurate modeling of the relationship between perturbations and generation quality, thereby addressing the issues you raised and potentially yielding further improvements.
>
> Our work is the first to model cache eviction as an optimization problem under perturbation constraints, meaning that **any refined perturbation metric can be seamlessly integrated into the proposed algorithm**, enabling future improvements.
>
> ## Reply to W2: Could you provide the comparisons between the greedy strategy and exhaustive search? Is it possible?
>
>
> **Exploring the global optimum via exhaustive search is computationally infeasible.** For example, consider a single-head attention layer with an input context length of 10,000, where we compress the cache to 2,000 tokens. The number of possible subsets is $ \binom{10000}{2000}\approx 10^{2170}$, which is astronomically large. In practical LLMs, the search space becomes even larger due to multiple layers and multiple heads.
> For this reason, a theoretically motivated greedy strategy is the only viable choice, whose effectiveness is validated by our empirical studies.
>
>
> ## Reply to W3: The conclusion of the output difference relies on attention weight, V, and W^O is straightforward. However, this article has spent a lot of space proving this, which makes me puzzled.
>
>
> Thank you for your question. While the high-level intuition, i.e., using attention weights, V, and $W^O$ to model output difference, may seem straightforward, **directly selecting tokens based on these information does not yield robust results.**
>
> So, our algorithm consists of two stages: the first stage uses attention weights for selection, and the second stage refines this by considering V and $W^O$. Stage 1 is crucial for safeguarding Theoretical Assumption 3.4. On this foundation, Stage 2 further limits output perturbations to optimize performance.
> As shown in Section 4.5 of our manuscript, removing Stage 1 (i.e., setting $\alpha=0$) may violate Assumption 3.4, leading to significant performance degradation. **The theoretical analysis is therefore necessary to explain why this degradation occurs and how the two-stage framework prevents it.**
>
> Overall, beyond simply proposing an effective algorithm, we prove that our method is theoretically grounded: it is equivalent to constraining an upper bound on the perturbation. This proof is essential for guaranteeing the algorithm's robustness.
>
> Thus, our algorithm is valuable in both theory and practice, offering stronger guarantees and improved empirical performance.
>
>
>
> ## Reply to W4: Why must this paper use 50% of the budget to select the KV pair with maximum attention weights? Why not use the adjusted attention containing output differences directly (In Alg. 1)?
>
>
>
> As noted above, **the first stage of our algorithm,** which selects top attention-weighted  cache pairs , **serves to safeguard Theoretical Assumption 3.4.** Directly using the adjusted attention (skipping Stage 1) violates this assumption. This theoretical risk is confirmed by our ablation study in Section 4.5: removing the first stage causes a substantial performance drop on Mistral-7B-v0.3 (from 42.85 to 31.94).
>
>
>
> Furthermore, the specific 50% budget is not rigid. As shown in the sensitivity analysis in Section 4.5, the algorithm is robust to this hyperparameter, consistently yielding strong performance with ratios ranging from 0.3 to 0.7.
>
> ═══════════════════════════════════════════════════════════════════════
>
> **We are grateful for your positive assessment, specifically your remark that “this paper is a good paper and the overall logic is self-motivated.” Your recognition means a lot to us. We hope that our response has effectively addressed your concerns, and would appreciate it if you could consider improving your score.**

---

> > ### Comment · Reviewer_NKA7 · 2025-11-23
> >
> > Thank you very much! This rebuttal has solved my concerns. I will take a look at other reviewers' opinions for the final decision.

---

> > > ### Author Response · Authors · 2025-11-25
> > >
> > > We sincerely thank you for your positive feedback and for confirming that our rebuttal has resolved your concerns. We will continue to engage with the other reviewers to clarify any remaining questions they may have.

---

### Official Review · Reviewer_fAVR · 2025-10-31

**Soundness:** 2
**Presentation:** 3
**Contribution:** 2
**Rating:** 4
**Confidence:** 4

**Summary:**

This paper studies KV cache eviction in LLM inference from an output perturbation perspective. To this end, the authors devised a perturbation-constrained objective to retain token caches that preserve the output distribution. In particular, given a total budget b, the authors divide b into two parts: the first part directly evicts caches with lower attention weights; the second part considers the L1 norm of value cache rows to identify important entries. Experiments demonstrate the effectiveness of the proposed method.

**Strengths:**

1. The proposed method is plug-and-play and easy to use.
2. The experimental results show clear improvements over baseline methods.

**Weaknesses:**

1. Some recent baselines are not compared [1, 2].
2. I believe the authors' definition of perturbation is too narrow—it is defined only based on the output at the current step, and the proposed method is built on this foundation. However, cache eviction is static. Although the perturbation may be reduced at the current step, the KV cache pruned based on this step's perturbation might become important again in subsequent steps. Therefore, how should the perturbation in future steps be measured? Also, how does the proposed method perform in multi-turn dialogue or ultra-long sequence scenarios?
3. The proposed method shares strong relevance to Quest [3]. Quest approximates attention computation based on the maximum and minimum KV values, while the proposed method uses the L2 norm for approximation. However, the authors do not provide any discussion or comparison with Quest, even though Quest employs a dynamic selection approach.
4. The comparison of inference speed is not sufficient. There should be results on acceleration for both 128K pre-filling and inference, similar to the presentation in DuoAttention [1].

[1] DuoAttention: Efficient Long-Context LLM Inference with Retrieval and Streaming Heads. In ICLR, 2025.
[2] PyramidKV: Dynamic KV Cache Compression based on Pyramidal Information Funneling. In Arxiv, 2024.
[3] Quest: Query-Aware Sparsity for Efficient Long-Context LLM Inference. In ICML, 2024.

**Questions:**

See weakness

---

> ### Author Response · Authors · 2025-11-22
> **Reply to reviewer (Part 1)**
>
> We appreciate your time and effort in reviewing our paper. **We are particularly grateful for your recognition of our method’s plug-and-play nature and clear improvements, which we believe are essential qualities for research in efficiency.**
>
>
> **We would first like to clarify the scope of our work.**  As detailed in the table below, Static Cache Eviction and Dynamic Cache Selection are orthogonal domains with distinct objectives and challenges. As stated in the Abstract and Introduction, **our contribution** is positioned as using formal, perturbation-based analysis to improve existing **Static Cache Eviction** methods. While the framework could in principle be extended, this is beyond the focus of the current paper and would require addressing setting-specific challenges. **We are open to adding “Cache Eviction” as a keyword in the title if this would help make the scope clearer.**
>
> | Category | Static Cache Eviction | Dynamic Cache Selection  |
> |-|-|-|
> |Representative Methods|**Ours**/AdaKV/SnapKV/HeadKV|Quest/MagicPig/MInference|
> |Operation Frequency| **Once** post-prefilling per layer to reduces peak memory.| **Per-step** dynamic selection for speedup. |
> | Per-Step Perturbation Modeling|**No.**  Infeasible for single-shot static compression. |**Yes.** Requires on-the-fly modeling at each step.|
> | Memory Saving | **Yes.** Permanently discards tokens, effectively reducing the VRAM footprint. | **No.** Typically retains the full cache in VRAM to support on-the-fly selection. |
> | Decoding Speedup | **Yes.** Accelerates attention computation via a reduced cache size.| **Yes.** Reduces latency by skipping computation and I/O for non-essential cache. |
> | Prefilling Speedup | **No.** Generally requires full computation (unless combined with chunked prefill). | **Yes.** Can sparsify computation directly during the prefilling phase. |
> |Primary Scope| 	**Memory-Constrained** Scenarios. Ideal for long-context serving on limited VRAM; enables larger batch sizes to greater throughput.| **Latency-Sensitive** Scenarios. Focuses on reducing inference latency by optimizing I/O and FLOPs, without reducing peak memory usage.|
>
>
>
> ═══════════════════════════════════════════════════════════════════════
>
> ## Reply to Weakness1：Some recent baselines are not compared
>
> Following your suggestion, we have incorporated PyramidKV and DuoAttention into our comparison. The table below presents the average scores across 16 datasets on LongBench.
>
> For instance, **DuoAttention** only achieves comparable performance (48.17) to our methods at 40% cache size on Llama, but **substantially underperforms in other settings—particularly on Mistral at 20% cache size (31.13), lagging behind ours methods (HeadKV w/ ours, 43.99) by over 12 points.**
>
> | Longbench Ave. Score (16 datasets) | Llama 20% | Llama 40% | Mistral 20% | Mistral 40% |
> |-|-|-|-|-|
> | Full Cache | 49.20 | 49.20 | 47.38 | 47.38 |
> | PyramidKV | 40.22 (18.3%↓) | 44.2 (10.2%↓) | 39.77 (16.1%↓) | 43.3 (8.6%↓) |
> | DuoAttention | 39.52 (19.7%↓) | **48.17 (2.1%↓)** | 31.13 (34.3%↓) | 43.74 (7.7%↓) |
> | AdaKV w/ ours | 43.77 (11.0%↓) | 48 (2.4%↓) | 42.85 (9.6%↓) | **46.23 (2.4%↓)** |
> | HeadKV w/ ours | **43.99 (10.6%↓)** | 47.79 (2.9%↓) | **43.46 (8.3%↓)** | 46.1 (2.7%↓) |
>
> We agree that including these baselines would be helpful and  **have added them in the revised version (Appendix B).**

---

> > ### Author Response · Authors · 2025-11-22
> > **Reply to Reviewer (Part 2)**
> >
> > ## Reply to W2: I believe the authors' definition of perturbation is too narrow.
> >
> > **Sketch (of Reply to W2).** **Our work focuses on static cache eviction, which is inherently limited in modeling perturbations at future steps.** Perturbation at future steps is fundamentally unobservable for such methods, and no prior cache eviction method models it. Our goal is to improve current-step perturbation under this inherent limitation of static eviction methods, not to compared to address dynamic cache selection.
> >
> >
> > Existing static cache eviction methods typically adopt heuristic strategies to handle attention variations across decoding steps: (1) measuring importance using the mean over multiple historical windows, and (2) setting moderate compression ratios (e.g., 40%–20%) to increase tolerance. **Our method inherits these established heuristic strategies from prior works**.
> >
> > **Although orthogonal dynamic cache selection methods such as Quest could model such perturbations, they have their own limitations**—most notably, they do not reduce the memory footprint. Thus, **demanding explicit modeling of future perturbations within the cache eviction paradigm is as unreasonable as requiring dynamic cache selection methods to save memory.** More importantly, these two lines of work can be naturally combined in practice. For example, one can first apply cache eviction to compress the KV cache to a moderate size (e.g., 50%), and then apply dynamic cache selection to further accelerate subsequent decoding.
> >
> > **Our goal is to replace the solely attention-based scoring mechanisms in existing cache eviction methods with a perturbation-constrained scoring algorithm.** In addition, our extensive experiments demonstrate clear improvements over a wide range of baselines. Although predicting future attention shifts may further boost performance, such techniques would be orthogonal and would also benefit our method.
> >
> > > Also, how does the proposed method perform in multi-turn dialogue or ultra-long sequence scenarios?
> >
> >
> > In response to your suggestion, we evaluated our method on the multi-turn dialogue benchmark SCBench. **SCBench contains 4,853 QA turns with an average length of 227K tokens, which not only far exceeds what academic research can typically afford, but also surpasses the context capacity of open-source LLMs.**
> >
> > To meet the constraints of 128K window of Llama-3.1-8B for multi-turn evaluation, we followed established practice and selected three representative tasks, Retr.KV, EN.QA, and Math.Find, from the three categories of String Retrieval, Semantic Retrieval, and Global Information, respectively, truncating the context to 100K tokens.
> >
> >
> > As shown in the table below, our method consistently yields improvements.
> >
> > | Avg. Score across multi turns | Full Cache | 80% Cache AdaKV | 80% Cache AdaKV (w/ ours) | 60% Cache AdaKV | 60% Cache AdaKV (w/ ours) | 40% Cache AdaKV | 40% Cache AdaKV (w/ ours) |
> > |-|-|-|-|-|-|-|-|
> > | Retr.KV (turns=5) | 52.2 | 52.00 | **52.6**   | 41.2 | **42**   | 19.4  | **19.8**   |
> > | Math.Find (turns=6) | 11.67 | 11.33 | **13.67** | 11.67 | **14.17** | 11.5 | **16.83** |
> > | EN.QA (turns=7) | 22.86 | 22.14| **24.29**  | 17.86 | **21.43**  | 15.71  | **22.14** |
> >
> > We have added this discussion in Section 4.4 of our revised version.
> >
> > ## Reply to W3: The proposed method shares strong relevance to Quest [3]. Quest approximates attention computation based on the maximum and minimum KV values.
> >
> >
> > **Quest belongs to the dynamic cache selection family**, **which approximates attention weights** to identify critical KV cache pages. Those methods, such as MagicPig, MInference and others, follow this paradigm and belong to this family. **We now discuss this family and its orthogonality to cache eviction in Appendix L.**
> >
> > Our work lies in a different static cache eviction family, where exact attention weights are available.  Guided by perturbation analysis, our method jointly leverages information from value vectors and pretrained parameters, achieving more accurate cache eviction than existing methods of this family.
> >
> > **It is also important to note that Quest does not use value vectors or pretrained parameter information, and its mechanism differs substantially from our pertubation-guided scoring.**
> >
> >
> > Extending our method to  dynamic cache selection settings like Quest is conceptually possible but challenging.  Dynamic methods must select critical tokens repeatedly at every decoding step. Conversely, static cache eviction operates infrequently—typically once post-prefilling—to permanently reduce memory usage. This structural difference explains why our perturbation-based selection is well-suited for static cache eviction but less applicable to dynamic selection, where per-step overhead is a critical constraint.
> >
> > Thus, a direct extension would therefore face efficiency challenges. We view this as an interesting direction for future work, **but beyond the scope of this paper.**

---

> > > ### Author Response · Authors · 2025-11-22
> > > **Reply to Reviewer (Part 3)**
> > >
> > > ## Reply to W4：The comparison of inference speed is not sufficient. There should be results similar to the presentation in DuoAttention.
> > >
> > > We have reported both pre-filling and decoding speeds for 16K–64K contexts in Figure 3. If the comment refers to acceleration for pre-filling using a chunked approach similar to DuoAttention, we would like to clarify that chunked pre-filling is a relatively straightforward and well-known technique.
> > >
> > > To address your concern, we conducted efficiency comparisons under the same conditions as DuoAttention. As shown below, our method incurs minimal overhead compared to AdaKV. Although DuoAttention achieves slightly faster TTFT due to its non-selective approach, it suffers from significantly lower quality, as previously discussed. Therefore, our method is preferable.
> > >
> > > In terms of decoding latency, DuoAttention’s official implementation requires separate passes for full attention and window attention. This results in higher latency than Full Cache at shorter context lengths (e.g., <60K). In contrast, **AdaKV and our method avoid this overhead, maintaining efficiency comparable to Full Cache at short contexts while achieving substantial acceleration (approx. 1.6x) at longer contexts.**
> > >
> > > 128K TTFT: Prefilling latency(s)
> > >
> > > | Chunk Size | Full Cache   | Duo Attention | AdaKV        | AdaKV (w/ ours) |
> > > |-|-|-|-|-|
> > > | 10k | 30.88(1.00x) | 20.66(1.49x)  | 21.57(1.43x) | 21.72(1.42x)    |
> > > | 30k | 30.57(1.00x) | 21.13(1.45x)  | 22.31(1.37x) | 22.48(1.36x)    |
> > > | 50k | 30.54(1.00x) | 21.97(1.39x)  | 23.46(1.30x) | 23.63(1.29x)    |
> > > | 70k | 30.50(1.00x) | 23.34(1.31x)  | 25.09(1.22x) | 25.24(1.21x)    |
> > > | 90k | 30.53(1.00x) | 25.09(1.22x)  | 25.99(1.17x) | 26.17(1.17x)    |
> > > | 110k | 30.43(1.00x) | 27.66(1.10x)  | 28.04(1.09x) | 28.21(1.08x)    |
> > > | 128k| 29.79(1.00x) | 29.81(1.00x)  | 29.98(0.99x) | 30.17(0.99x)    |
> > >
> > > Decoding latency(ms)
> > >
> > > | Context Len | Full Cache    | Duo Attention | AdaKV         | AdaKV (w/ ours) |
> > > |-|-|-|-|-|
> > > | 20K | 30.90 (1.00x) | 39.55 (0.78x) | 30.97 (1.00x) | 31.80 (0.97x)   |
> > > | 40K | 30.97 (1.00x) | 39.65 (0.78x) | 31.10 (1.00x) | 31.13 (0.99x)   |
> > > | 60K | 35.14 (1.00x) | 39.76 (0.88x) | 30.95 (1.14x) | 31.04 (1.13x)   |
> > > | 80K | 41.39 (1.00x) | 39.38 (1.05x) | 31.31 (1.32x) | 31.10 (1.33x)   |
> > > | 100K | 47.73 (1.00x) | 40.09 (1.19x) | 32.22 (1.48x) | 31.77 (1.50x)   |
> > > | 120K | 54.18 (1.00x) | 39.86 (1.36x) | 34.88 (1.55x) | 34.92 (1.55x)   |
> > > | 140K | 60.69 (1.00x) | 40.37 (1.50x) | 38.05 (1.60x) | 38.06 (1.59x)   |
> > >
> > > ═══════════════════════════════════════════════════════════════════════
> > >
> > > ## Final Remarks
> > >
> > > **We have performed comprehensive experiments to address your concerns. If you have any further questions, please do not hesitate to let us know and we are happy to discuss them further. If these results have alleviated your concerns, we would greatly appreciate if you could consider raising your score.**
> > >
> > > **Thank you again for your time and effort dedicated to this paper.**

---

### Author Response · Authors · 2025-12-02
**Final Remarks**

Dear Reviewers, ACs, SACs, and PCs,

We thank all reviewers for their time and valuable feedback.

We especially appreciate the positive assessments from Reviewers NKA7 and FWKD, who highlighted **our theoretically grounded, self-motivated analysis of KV-cache criticality, our algorithm's plug-and-play compatibility with SOTA methods, and our extensive validation across model families with reducing compression loss by more than half**.

However, we believe that **the similar negative comments** from Reviewers fAVR and FGK5 **stem from conflating Static Cache Eviction with Dynamic Cache Selection**.    As detailed in the table below, **these domains are orthogonal**. Our work focuses explicitly on using formal perturbation analysis to improve Static Cache Eviction. While our framework could theoretically extend to other settings, addressing the specific challenges of dynamic selection is **beyond the scope of this paper.**

Below we clarify **the same misunderstandings from Reviewers fAVR and FGK5**:

1. **On a factual misunderstanding of Quest’s use of value states.**  **Both reviewers fAVR and FGK5 misunderstand Quest's mechanism.** As a Dynamic Cache Selection method, Quest **only approximates attention weights using the maximum and minimum key vectors; it does not utilize value vectors**. Our approach is **fundamentally different both technically (utilizing both value vectors and pretrained parameters for analysis)** and in its domain application.

2. **On per-step perturbation as a core flaw of our method.**  Reviewers suggested that our static cache eviciton method failing to model per-step perturbation is a core flaw.
However, this is **an inherent limitation of all static cache eviction approaches.** Existing methods typically handle cross-step variation using established heuristics: (a) measuring averaged importance over windows, and (b) using moderate compression ratios for robustness. **Our method explicitly inherits these established heuristics from prior work.** While dynamic cache selection approaches model these perturbations, they do not reduce memory usage. **Requiring explicit modeling of future perturbations within static cache eviction is as unreasonable as demanding memory reduction from dynamic selection methods.** Additionally, these two domains are **complementary in practice**: static eviction is first used to reduce cache memory usage, and dynamic selection is then applied for further speedup.

3. **On cache eviction speed evaluation and the role of chunk prefill.**  Both reviewers questioned our speed measurements, **despite our strict adherence to common evaluation protocols in this field**. Furthermore, regarding cache eviction with chunk prefill, it is widely recognized to degrade generation quality and is rarely evaluated. To our knowledge, almost no prior work reports quality metrics under chunk prefill;  even the cited DuoAttention reports only speed, without quality. Nevertheless, **following reviewer requests, we have provided additional efficiency results demonstrating our method’s competitive speed**.

In addition, we have conducted further experiments requested by both reviewers fAVR and FGK5, including: (a.) adding more baselines, and  (b.) evaluating on the multi-turn dialogue benchmark SCBench.  Our method demonstrates **clear effectiveness in all these new settings.**

Regrettably, **neither Reviewer fAVR nor FGK5 responded during the discussion phase**, preventing us from further clarifying. We believe our paper makes a meaningful contribution to the static cache eviction literature. We hope the above clarifications assist the AC in making the final decision, and we look forward to that decision.

Best regards,
Authors

| Category | Static Cache Eviction | Dynamic Cache Selection  |
|-|-|-|
|Representative Methods|**Ours**/AdaKV/SnapKV/HeadKV|Quest/MagicPig/MInference|
|Operation Frequency| **Once** post-prefilling per layer to reduces peak memory.| **Per-step** dynamic selection for speedup. |
| Per-Step Perturbation Modeling|**No.**  Infeasible for single-shot static compression. |**Yes.** Requires on-the-fly modeling at each step.|
| Memory Saving | **Yes.** Permanently discards tokens, effectively reducing the VRAM footprint. | **No.** Typically retains the full cache in VRAM to support on-the-fly selection. |
| Decoding Speedup | **Yes.** Accelerates attention computation via a reduced cache size.| **Yes.** Reduces latency by skipping computation and I/O for non-essential cache. |
| Prefilling Speedup | **No.** Generally requires full computation (unless combined with chunked prefill). | **Yes.** Can sparsify computation directly during the prefilling phase. |
|Primary Scope| 	**Memory-Constrained** Scenarios. Ideal for long-context serving on limited VRAM; enables larger batch sizes to greater throughput.| **Latency-Sensitive** Scenarios. Focuses on reducing inference latency by optimizing I/O and FLOPs, without reducing peak memory usage.|

---

### Meta-Review · Area_Chair_p7U1 · 2026-01-07

**Summary:**

The paper targets static KV-cache eviction via output-perturbation analysis and proposes a two-stage, plug-and-play selection scheme. Empirical results on LongBench and Ruler show consistent gains over several eviction baselines. However, key concerns remain. The positioning versus dynamic selection methods is only partially convincing; novelty beyond an intuitively output-aware view feels incremental. Evaluation emphasizes single-turn prompts and adds multi-turn SCBench only in rebuttal with limited scope. Efficiency analysis relies heavily on TTFT; per-step overhead and end-to-end throughput under realistic serving regimes are not synthesized in the main text. Presentation is verbose, and the theory-to-algorithm pipeline could be clearer.

**Reviewer Concerns:**

The rebuttal helps by clarifying scope (static eviction vs dynamic selection), adding recent baselines (PyramidKV, DuoAttention), reporting SCBench results, showing chunked-prefill latencies, and committing to clearer naming (CriticalKV) and streamlined exposition. It also argues the infeasibility of exhaustive search and provides α-sensitivity, including at 10% cache.

Still open are issues central to acceptance. The claimed “formal” advance reduces to a theory-guided heuristic that aligns with a straightforward output-aware intuition; the novelty relative to prior output-sensitive or selection-based lines remains thin. The evaluation breadth for multi-turn and ultra-long contexts is limited and presented late. Efficiency reporting is fragmented and omits a concise accounting of overheads per step and throughput across context lengths in the main body. The method’s dependence on a two-stage design and a fixed split (α) would benefit from stronger justification and sensitivity in the main text.

**Reviewer Scores:**

R1 (4): Added baselines, SCBench, and efficiency details partially address concerns, but skepticism about scope and metrics persists.
R2 (2): Clarifications and added results improve confidence, yet reservations about formality and suitability remain.
R3 (6): Unchanged. Positive but notes lingering questions on clarity and necessity of the two-stage design.
R4 (8): Unchanged. Their specific requests (α sensitivity) were addressed; broader novelty concerns lie outside their critique. Short review

---

### Decision · Program_Chairs · 2026-01-26

Reject